# Sigmoid Gating is More Sample Efficient than Softmax Gating in Mixture of Experts

Huy Nguyen    Nhat Ho$^\star$    Alessandro Rinaldo$^\star$

Department of Statistics and Data Sciences, The University of Texas at Austin$^\dagger$

{huynm, minhnhat}@utexas.edu, alessandro.rinaldo@austin.utexas.edu

## Abstract

The softmax gating function is arguably the most popular choice in mixture of experts modeling. Despite its widespread use in practice, the softmax gating may lead to unnecessary competition among experts, potentially causing the undesirable phenomenon of representation collapse due to its inherent structure. In response, the sigmoid gating function has been recently proposed as an alternative and has been demonstrated empirically to achieve superior performance. However, a rigorous examination of the sigmoid gating function is lacking in current literature. In this paper, we verify theoretically that the sigmoid gating, in fact, enjoys a higher sample efficiency than the softmax gating for the statistical task of expert estimation. Towards that goal, we consider a regression framework in which the unknown regression function is modeled as a mixture of experts, and study the rates of convergence of the least squares estimator under the over-specified case in which the number of fitted experts is larger than the true value. We show that two gating regimes naturally arise and, in each of them, we formulate an identifiability condition for the expert functions and derive the corresponding convergence rates. In both cases, we find that experts formulated as feed-forward networks with commonly used activation such as $\mathrm{ReLU}$ and $\mathrm{GELU}$ enjoy faster convergence rates under the sigmoid gating than those under softmax gating. Furthermore, given the same choice of experts, we demonstrate that the sigmoid gating function requires a smaller sample size than its softmax counterpart to attain the same error of expert estimation and, therefore, is more sample efficient.

## 1 Introduction

Mixture of experts (MoE) [15, 17] has recently emerged as a powerful machine learning model that helps scale up the model capacity while requiring a nearly constant computational overhead [34, 8]. In particular, it aggregates multiple sub-models called experts based on a gating network. Here, experts can be formulated as neural networks, and they specialize in different aspects of the data. For instance, in large language models [16, 31, 19, 40, 7, 5, 30], one expert might focus on syntax, another on semantics, whereas the others concentrate on context. Meanwhile, the gating network takes the input data and calculates a set of weights that determine the contribution of each expert. This gating mechanism guarantees that the most relevant experts are assigned more weight for a given input. In addition to language modeling, MoE has also been leveraged in several other applications, including computer vision [32, 20, 33], multi-task learning [12, 10], multi-modal learning [11], speech recognition [38, 9, 29] and reinforcement learning [3, 28].

---

$^\star$Co-last authors.

In the above applications of MoE, practitioners frequently utilize the softmax gating function to compute the mixture weights. However, the use of softmax function might introduce an unexpected competition among experts, which leads to the representation collapse issue [2, 30, 18, 24]: when the weight of one expert increases, those of the others decrease accordingly. As a consequence, some experts will dominate the decision-making process, and overshadow the contributions of others. Therefore, the ability of MoE models to incorporate the diversity of expert specialization is partially limited. To this end, Csordás et al. [4] propose using the sigmoid function in place of the softmax to remove the unnecessary expert competition, and empirically demonstrate that it is indeed a better choice of gating function. On the other hand, the theoretical guarantee for that claim has remained missing in the literature. The main objective of this work is to provide a comprehensive analysis of the sigmoid gating function from the perspective of the expert estimation problem. In particular, we will show that sigmoid gating delivers superior sample efficiency for estimating the model parameters and allows for more general expert functions than softmax gating.

**Related works.** A very recent line of research has focused on analyzing the convergence rates for expert estimation in Gaussian MoE models under a variety of assumptions and choices of gating functions. Assuming that data from an input-free gating Gaussian MoE, Ho et al. [14] demonstrated that the expert estimation rates for the maximum likelihood estimator vary with the algebraic independence among expert functions. Under the same setting but using instead softmax gating, Nguyen et al. [27] discovered that the expert estimation rates depend on the solvability of a system of polynomial equations resulting from an intrinsic interaction among gating and expert parameters. Subsequently, a Gaussian MoE with Top-K sparse softmax gate, designed to activate only a fraction of experts per input, was studied in [25]. The convergence analysis in that work revealed that activating exactly one expert for each input would eliminate the previous parameter interaction, thus boosting the expert estimation rates. Despite many important insights provided in this line of works, the mixture setting of Gaussian MoE is still far from practical. To that effect, Nguyen et al. [26] introduced a more realistic regression framework in which, conditionally on the features, the response variables are sampled from noisy realization of an unknown and deterministic softmax gating MoE-type regression function. They found that experts formulated as feed-forward networks with sigmoid or hyperbolic tangent activation functions enjoy faster estimation rates than those equipped with a polynomial activation. In this paper, we adopt the same regression setting and carry out an analogous analysis by considering instead sigmoid gating for the underlying regression function, as opposed to the more popular softmax gating considered in [26]. As we will see, the choice of gating function turns out to be critical, requiring a different type of analysis and leading to markedly different conclusions.

**Problem setup.** We assume that the data $(X_1, Y_1), (X_2, Y_2), \ldots, (X_n, Y_n) \in \mathbb{R}^d \times \mathbb{R}$ follow a standard regression model

$$Y_i = f_{G_*}(X_i) + \epsilon_i, \quad i = 1, \ldots, n, \tag{1}$$

where the features $X_1, X_2, \ldots, X_n$ are i.i.d. samples from a probability distribution $\mu$ on $\mathbb{R}^d$ and $\epsilon_1, \ldots, \epsilon_n$ are independent noise variables such that $\mathbb{E}[\epsilon_i | X_i] = 0$ and $\text{Var}(\epsilon_i | X_i) = \nu$ for all $1 \leq i \leq n$; for simplicity we further assume that they follow a Gaussian distribution. The unknown regression function $f_{G_*}$ is formulated as a sigmoid-gated MoE with $k_*$ experts, i.e.,

$$x \in \mathbb{R}^d \mapsto f_{G_*}(x) := \sum_{i=1}^{k_*} \frac{1}{1 + \exp(-(\beta_{1i}^*)^\top x - \beta_{0i}^*)} \cdot h(x, \eta_i^*), \tag{2}$$

where the *expert function* $x \mapsto h(x, \eta)$ is of parametric form, specified by parameter $\eta \in \mathbb{R}^q$. The regression function $f_{G_*}$ is fully characterized by the unknown parameters $(\beta_{0i}^*, \beta_{1i}^*, \eta_i^*)_{i=1}^{k_*}$ in $\mathbb{R} \times \mathbb{R}^d \times \mathbb{R}^q$, which can be compactly encoded using the associated *mixing measure* $G_* := \sum_{i=1}^{k_*} \frac{1}{1+\exp(-\beta_{0i}^*)} \delta_{(\beta_{1i}^*, \eta_i^*)}$, a weighted sum of Dirac measures $\delta$.

**Least squares estimation.** To estimate the unknown parameters $(\beta_{0i}^*, \beta_{1i}^*, \eta_i^*)_{i=1}^{k_*}$ (equivalently, the ground truth mixing measure $G^*$), we deploy the least squares method [35] and focus on the estimator

$$\widehat{G}_n := \underset{G \in \mathcal{G}_k(\Theta)}{\arg\min} \sum_{i=1}^{n} \left(Y_i - f_G(X_i)\right)^2, \tag{3}$$

where $\mathcal{G}_k(\Theta) := \{ G = \sum_{i=1}^{k'} \frac{1}{1+\exp(-\beta_{0i})} \delta_{(\beta_{1i}, \eta_i)} : 1 \le k' \le k, \ (\beta_{0i}, \beta_{1i}, \eta_i) \in \Theta \}$ is the set of all mixing measures with at most $k$ atoms. Since the true number of true experts $k_*$ is typically unknown in practice, we will assume that the number $k$ of fitted experts is at least as large, i.e. that $k > k_*$.

**Technical challenges.** The main challenge in analyzing MoE models with sigmoid gating lies in the convergence behavior of the mixture weights. Specfically, since we fit the ground-truth MoE model (2) with a mixture of $k > k_*$, there must be some true atoms $(\beta_{1i}^*, \eta_i^*)$ fitted by more than one component; we will refer to the corresponding parameters $\beta_{1i}^*$ as *over-specified parameters*. To illustrate, suppose that $(\hat{\beta}_{1i}^n, \hat{\eta}_i^n) \to (\beta_{11}^*, \eta_1^*)$ for $i \in \{1, 2\}$, in probability. Then, to ensure convergence of $f_{\widehat{G}_n}$ to $f_{G_*}$ in the $L^2(\mu)$-norm we must have that, for $i = 1, 2$,

$$ \sum_{i=1}^2 \frac{1}{1 + \exp(-(\hat{\beta}_{1i}^n)^\top x - \hat{\beta}_{0i}^n)} \to \frac{1}{1 + \exp(-(\beta_{11}^*)^\top x - \beta_{01}^*)}, $$

as $n \to \infty$, for $\mu$-almost every $x$. Note that the above limit can be achieved only if $\beta_{11}^* = 0_d$. As a result, we will consider the two following regimes for the over-specified parameters $\beta_{1i}^*$:

**Regime 1.** All the over-specified parameters $\beta_{1i}^*$ are equal to $0_d$;

**Regime 2.** At least one among the over-specified parameters $\beta_{1i}^*$ is different from $0_d$.

It is worth emphasizing that the second regime presents additional technical challenges, as the least squares estimator $\widehat{G}_n$ converges to a mixing measure $\overline{G} \in \arg\min_{G \in \mathcal{G}_k(\Theta) \setminus \mathcal{G}_{k_*}(\Theta)} \|f_G - f_{G_*}\|_{L^2(\mu)}$ that is in general different that the true mixing measure $G_*$, as in the first regime.

**Contributions.** In this paper, we carry out a convergence analysis of the sigmoid gating MoE under two regimes of the gating parameters. Our main objective is to compare the sample efficiency between the sigmoid gating and the softmax gating. The contributions of our paper are three-fold, and can be summarized as follows:

**(C.1) Convergence rate for the regression function.** We demonstrate in Theorem 1 that the regression estimation $f_{\widehat{G}_n}$ converges to its true counterpart $f_{G_*}$ at the rate of order $\mathcal{O}_P(n^{-1/2})$, which is parametric on the sample size $n$. This regression estimation rate is then utilized for determining the expert estimation rates.

**(C.2) Expert estimation rates under the Regime 1.** Under the first regime, we first establish a condition called *strong identifiability* to characterize which types of experts would yield polynomial estimation rates. In particular, we find out that the rates for estimating experts formulated as feed-forward networks with popular activation functions such as ReLU and GELU are of polynomial orders. By contrast, those for polynomial experts and input-indepedent experts are slower than any polynomial rates and could be of order $\mathcal{O}_P(1/\log(n))$. Such expert convergence behavior is similar to that when using the softmax gating.

**(C.3) Expert estimation rates under the Regime 2.** Under the second regime, the regression estimation $f_{\widehat{G}_n}$ converge to a function taking the form of a sigmoid gating MoE which is different from $f_{G_*}$. From our derived *weak identifiability* condition, it follows that estimation rates for feed-forward expert networks with ReLU or GELU activation and polynomial experts are of orders $O_P(n^{-1/2})$, which are substantially faster than those when using the softmax gating (see also Table 1). Therefore, it follows that the sigmoid gating is more sample efficient than the softmax gating.

**Outline.** The rest of the paper is organized as follows. In Section 2, we introduce some mild assumptions on the parameters, and then determine estimation rates for the regression function. Next, we establish convergence rates for expert estimation under both Regime 1 and the Regime 2 in Section 3. Then, we conduct some experiments to empirically justify our theory in Section 4. Finally, we provide a discussion on the practical implications, limitations and future directions of our work in Section 5. Full proofs of the results and additional experiments are deferred to the Appendices.

**Notation.** For any $n \in \mathbb{N}$, we denote $[n]$ as the set $= \{1, 2, \ldots, n\}$. Additionally, for any set $S$, we refer to $|S|$ as its cardinality. Next, for any vectors $v := (v_1, v_2, \ldots, v_d) \in \mathbb{R}^d$ and $\alpha := (\alpha_1, \alpha_2, \ldots, \alpha_d) \in \mathbb{N}^d$, we let $v^\alpha = v_1^{\alpha_1} v_2^{\alpha_2} \ldots v_d^{\alpha_d}$, $|v| := v_1 + v_2 + \ldots + v_d$ and $\alpha! := \alpha_1! \alpha_2! \ldots \alpha_d!$, while $\|v\|$ stands for its 2-norm value. Lastly, for any two positive sequences $(a_n)_{n \ge 1}$ and $(b_n)_{n \ge 1}$, we write $a_n = \mathcal{O}(b_n)$ or $a_n \lesssim b_n$ if $a_n \le C b_n$ for all $n \in \mathbb{N}$, where $C > 0$ is some universal constant. The notation $a_n = \mathcal{O}_P(b_n)$ indicates that $a_n/b_n$ is stochastically bounded.

Table 1: Summary of expert estimation rates (up to a logarithmic factor) under the MoE models equipped with the sigmoid gating (ours) and the softmax gating [26]. In this work, we consider three types of expert functions including experts network with ReLU, GELU activations; polynomial experts; and input-independent experts.

| | ReLU, GELU Experts | | Polynomial Experts | | Input-independent Experts |
|---|---|---|---|---|---|
| | Regime 1 | Regime 2 | Regime 1 | Regime 2 | |
| **Sigmoid** | $\mathcal{O}_P(n^{-1/4})$ | $\mathcal{O}_P(n^{-1/2})$ | $\mathcal{O}_P(1/\log(n))$ | $\mathcal{O}_P(n^{-1/2})$ | $\mathcal{O}_P(1/\log(n))$ |
| **Softmax** | $\mathcal{O}_P(n^{-1/4})$ | | $\mathcal{O}_P(1/\log(n))$ | | $\mathcal{O}_P(1/\log(n))$ |

## 2 Preliminaries

In this section, we demonstrate the parametric convergence rate for the least squares estimator of the regression function $f_{\widehat{G}_n}$. We begin by describing the assumption that will be used throughout the paper, which are overall very mild.

**(A.1) Topological assumptions.** To ensure the convergence of the least squares estimators, we assume the parameter space $\Theta \subseteq \mathbb{R} \times \mathbb{R}^d \times \mathbb{R}^q$ is compact, and the input space $\mathcal{X} \subseteq \mathbb{R}^d$ is bounded.

**(A.2) Lipschitz experts.** The expert function $h(\cdot, \eta)$ is assumed to be Lipschitz continuous with respect to $\eta$ and bounded, $\mu$-almost surely.

**(A.3) Distinct experts.** The ground-truth expert parameters $\eta_1^*, \ldots, \eta_{k_*}^*$ are pair-wise different so that the experts $h(x, \eta_1^*), \ldots, h(x, \eta_{k_*}^*)$ are also distinct from each other.

Given the above assumptions, we are now ready to present the main result of this section. Recall that we divide our analysis into two following regimes:

- **Regime 1.** All the over-specified parameters $\beta_{1i}^*$ are equal to $0_d$;
- **Regime 2.** At least one among the over-specified parameters $\beta_{1i}^*$ is different from $0_d$.

Let us begin with Regime 1. We show in Theorem 1 that the regression estimator $f_{\widehat{G}_n}$ converges to the true regression function $f_{G_*}$ at the parametric rate.

**Theorem 1.** *Under the Regime 1 and with the least squares estimator $\widehat{G}_n$ defined in equation* (3)*, the regression estimator $f_{\widehat{G}_n}$ admits the following rate of convergence to $f_{G_*}$:*

$$\|f_{\widehat{G}_n} - f_{G_*}\|_{L^2(\mu)} = \mathcal{O}_P([\log(n)/n]^{\frac{1}{2}}). \tag{4}$$

The proof of Theorem 1 is in Appendix A.1. The above result indicates that if we are able to design a loss function among parameters $\mathcal{D}$ that satisfies the lower bound $\|f_{\widehat{G}_n} - f_{G_*}\|_{L^2(\mu)} \gtrsim \mathcal{D}(\widehat{G}_n, G_*)$, then we obtain parameter estimation rates via the bound $\mathcal{D}(\widehat{G}_n, G_*) = \mathcal{O}_P([\log(n)/n]^{\frac{1}{2}})$. Consequently, those parameter estimation rates lead to our desired expert estimation rates.

Next, under the Regime 2, the true model is misspecified, i.e., the regression estimator $f_{\widehat{G}_n}$ converges to the missepcified regression function $f_{\overline{G}}$ rather than the true regression function $f_{G_*}$, where $\overline{G} \in \overline{\mathcal{G}}_k(\Theta) := \arg\min_{G \in \mathcal{G}_k(\Theta) \setminus \mathcal{G}_{k_*}(\Theta)} \|f_G - f_{G_*}\|_{L^2(\mu)}$. By employing the same arguments as in Theorem 1, we also capture the parametric regression estimation rate $\mathcal{O}_P([\log(n)/n]^{\frac{1}{2}})$ for $\|f_{\widehat{G}_n} - f_{\overline{G}}\|_{L^2(\mu)}$ under this regime, which is stated in the following corollary.

**Corollary 1.** *Under the Regime 2, the regression estimator $f_{\widehat{G}_n}$ admits the following rate of convergence to $f_{\overline{G}}$:* $\inf_{\overline{G} \in \overline{\mathcal{G}}_k(\Theta)} \|f_{\widehat{G}_n} - f_{\overline{G}}\|_{L^2(\mu)} = \mathcal{O}_P([\log(n)/n]^{\frac{1}{2}})$.

## 3 Convergence Rates for Expert Estimation

In this section we will establish the expert estimation rates under both Regime 1 and the Regime 2; see Section 3.1 and Section 3.2, respectively. Under each regime, we will formulate appropriate

conditions, which we refer to as *identifiability conditions*, on the expert functions that will guarantee fast estimation rates. Furthermore, we will determine the (slow) estimation rates of some commonly used experts which fail to satisfy such identifiability conditions.

## 3.1 Regime 1 of Gating Parameters

Recall that under the Regime 1, all the over-specified parameters $\beta_{1i}^*$, i.e., those fitted by at least two parameters, are equal to $0_d$. Without loss of generality, we assume that $\beta_{11}^*, \ldots, \beta_{1\bar{k}}^*$ are over-specified parameters, where $1 \leq \bar{k} \leq k_*$. The remaining parameters $\beta_{11}^* = \ldots = \beta_{1\bar{k}}^* = 0_d$. Meanwhile, $\beta_{1(\bar{k}+1)}^*, \ldots, \beta_{1k_*}^*$ are assumed to be fitted by exactly one estimator.

Following a strategy that has been successfully used for analyzing MoE and mixture models (see, e.g. [21] and [26]), in order to derive the expert estimation rates, it is sufficient to propose a Voronoi loss function $\mathcal{D}$ among parameters, and then prove the lower bound $\|f_{\widehat{G}_n} - f_{G_*}\|_{L^2(\mu)} \gtrsim \mathcal{D}(\widehat{G}_n, G_*)$. Given the parametric regression estimation rate in Theorem 1, we then obtain that $\mathcal{D}(\widehat{G}_n, G_*) = \mathcal{O}_P([\log(n)/n]^{\frac{1}{2}})$. Note that a key step in deriving the previous lower bound is to decompose the difference $f_{\widehat{G}_n}(x) - f_{G_*}(x)$ into a combination of linearly independent terms.

Towards that goal, we will Taylor expand to the product of mixture weight and expert functions $F(x, \beta_1, \beta_0, \eta) := \sigma(x, \beta_1, \beta_0)h(x, \eta)$, where we let $\sigma(x, \beta_1, \beta_0) := \frac{1}{1+\exp(-\beta_1^\top x - \beta_0)}$. To obtain the desired decomposition of $f_{\widehat{G}_n}(x) - f_{G_*}(x)$, we provide below in Definition 1 a condition, which we refer to as a *strong identifiability* condition, that guarantees that the derivatives of the product $F(x, \beta_1, \beta_0, \eta)$ w.r.t its parameters are linearly independent.

**Definition 1** (Strong identifiability). *An expert function $h(x, \eta)$ is called strongly identifiable if it is twice differentiable w.r.t its parameter $\eta$ for $\mu$-almost all $x$ and, for any positive integer $\ell$ and any pair-wise distinct choices of parameters $\{(\beta_{0i}, \beta_{1i}, \eta_i)\}_{i=1}^\ell$, the functions in the classes*

$$\left\{ x \mapsto \frac{\partial^{|\gamma_1|+|\gamma_2|} F}{\partial \beta_1^{\gamma_1} \partial \eta^{\gamma_2}}(x, 0_d, \beta_{0i}, \eta_i) : i \in [\ell], (\gamma_1, \gamma_2) \in \mathbb{N}^d \times \mathbb{N}^q, 1 \leq |\gamma_1| + |\gamma_2| \leq 2 \right\}$$

*and*

$$\left\{ x \mapsto \frac{\partial F}{\partial \beta_1^{\alpha_1} \partial \beta_0^{\alpha_2} \partial \eta^{\alpha_3}}(x, \beta_{1i}, \beta_{0i}, \eta_i) : i \in [\ell], (\alpha_1, \alpha_2, \alpha_3) \in \mathbb{N}^d \times \mathbb{N} \times \mathbb{N}^q, |\alpha_1| + \alpha_2 + |\alpha_3| = 1 \right\}$$

*are linearly independent, for $\mu$-almost all $x$.*

**Example.** Consider experts formulated as two-layer neural networks, i.e., $h(x, (a, b)) = \varphi(a^\top x + b)$, where $\varphi$ is an activation function and $(a, b) \in \mathbb{R}^d \times \mathbb{R}$. If $a = 0_d$, then the expert $h(\cdot, (a, b))$ is not strongly identifiable for any choices of the activation function $\varphi$. On the other hand, if $a \neq 0_d$ and $\varphi$ is one among popular activation functions such as ReLU and GELU, then $h(\cdot, (a, b))$ is a strongly identifiable expert. By contrast, if $\varphi$ is a polynomial, e.g. $\varphi(z) = z^p$, then the expert $h(\cdot, (a, b))$ does not meet the strong identifiability condition.

Intuitively, the strong identifiability condition helps eliminate potential interactions among parameters expressed in the language of PDE (see equations (8) and (11) where gating parameters $\beta_1$ interact with expert parameters $a$). Such interactions are demonstrated to result in significantly slow expert estimation rates presented in Theorem 3 and 4). From the technical perspective, a key step in our proof techniques rely on the decomposition of the discrepancy $f_{\widehat{G}_n}(x) - f_{G_*}(x)$ into a combination of linearly independent terms. This can be done by applying Taylor expansions to the function $F(x, \beta_1, \beta_0, \eta) = \sigma(\beta_1^\top x + \beta_0)h(x, \eta)$ defined as the product of the sigmoid gating and the expert function $h$. Thus, the condition is to ensure that terms in the decomposition are linearly independent.

Next, following the strategy first introduced by Manole et al. [21] and then adopted in several convergence analyses of MoE [22, 23], we proceed to construct a loss function among parameters based on the notion of Voronoi cells.

**Voronoi loss.** Given an arbitrary mixing measure $G$ with $k' \leq k$ atoms, we distribute its atoms across the Voronoi cells $\{\mathcal{A}_j \equiv \mathcal{A}_j(G), j \in [k_*]\}$ generated by the atoms of $G_*$, where

$$\mathcal{A}_j := \{i \in [k'] : \|\omega_i - \omega_j^*\| \leq \|\omega_i - \omega_\ell^*\|, \forall \ell \neq j\}, \tag{5}$$

with $\omega_i := (\beta_{1i}, \eta_i)$ and $\omega_j^* := (\beta_{1j}^*, \eta_j^*)$. In particular, the cardinality of the Voronoi cell $\mathcal{A}_j$ corresponding to the least squares estimator $\widehat{G}_n$ in (3) is the number of fitted components assigned (and, likely, converging) to the true atoms $\omega_j^*$. Then, we define the Voronoi loss function

$$\mathcal{D}_1(G, G_*) := \sum_{j=1}^{\bar{k}} \Big| \sum_{i \in \mathcal{A}_j} \frac{1}{1 + \exp(-\beta_{0i})} - \frac{1}{1 + \exp(-\beta_{0j}^*)} \Big| + \sum_{j=1}^{\bar{k}} \sum_{i \in \mathcal{A}_j} \Big[ \|\Delta\beta_{1ij}\|^2 + \|\Delta\eta_{ij}\|^2 \Big]$$
$$+ \sum_{j=\bar{k}+1}^{k_*} \sum_{i \in \mathcal{A}_j} \Big[ \|\Delta\beta_{1ij}\| + |\Delta\beta_{0ij}| + \|\Delta\eta_{ij}\| \Big], \quad (6)$$

where we let $\Delta\beta_{1ij} := \beta_{1i} - \beta_{1j}^*$, $\Delta\beta_{0ij} := \beta_{0i} - \beta_{0j}^*$, and $\Delta\eta_{ij} := \eta_i - \eta_j^*$. Above, if the Voronoi cell $\mathcal{A}_j$ is empty, then the summation term becomes zero by convention. Additionally, since $\mathcal{D}_1(G, G_*) = 0$ if and only if $G \equiv G_*$, it follows that when $\mathcal{D}_1(G, G_*)$ is sufficiently small, the differences $\Delta\beta_{0ij}$, $\Delta\beta_{1ij}$ and $\Delta\eta_{ij}$ are also small. This observation suggests that, though not symmetric in its arguments and therefore not a metric, $\mathcal{D}_1(G, G_*)$ is nonetheless a suitable loss function for capturing the discrepancy between the least squares estimator $\widehat{G}_n$ and the true mixing measures $G_*$. Furthermore, it is worth noting that the Voronoi loss function $\mathcal{D}_1$ can be efficiently evaluated, as its computational complexity is of order $\mathcal{O}(k \times k_*)$.

In our next result, whose proof can be found in Appendix A.2, we show that the Voronoi loss between the least squares estimator and the true mixing measure is upper bounded, up to universal multiplicative constants, by the estimation error for the regression function.

**Theorem 2.** *Let $x \mapsto h(x, \eta)$ be a strongly identifiable expert function. Then the lower bound*

$$\|f_G - f_{G_*}\|_{L^2(\mu)} \gtrsim \mathcal{D}_1(G, G_*),$$

*holds true for any $G \in \mathcal{G}_k(\Theta)$. As a consequence, this result together with the regression estimation rate in Theorem 1 indicate that $\mathcal{D}_1(\widehat{G}_n, G_*) = \mathcal{O}_P([\log(n)/n]^{\frac{1}{2}})$.*

The following remarks regarding the results of Theorem 2 are in order.

**(i)** It follows from the formulation of the loss function $\mathcal{D}_1$ that the estimation rates for the over-specified parameters $\beta_{1j}^*, \eta_{1j}^*$, where $j \in [\bar{k}]$, are all of order $\mathcal{O}_P([\log(n)/n]^{\frac{1}{4}})$. Note that as the expert $h(\cdot, \eta)$ is twice differentiable over a bounded domain, it is also a Lipschitz function. Thus, letting $\widehat{G}_n := \sum_{i=1}^{\widehat{k}_n} \frac{1}{1+\exp(-\widehat{\beta}_{0i})} \delta_{(\widehat{\beta}_{1i}^n, \widehat{\eta}_i^n)}$, we obtain that

$$\sup_x |h(x, \widehat{\eta}_i^n) - h(x, \eta_j^*)| \le L_1 \|\widehat{\eta}_i^n - \eta_j^*\| \lesssim \mathcal{O}_P([\log(n)/n]^{\frac{1}{4}}), \quad (7)$$

for any $i \in \mathcal{A}_j(\widehat{G}_n)$, where $L_1 \ge 0$ is a Lipschitz constant. The above bound indicates that if the strongly identifiable expert $h(\cdot, \eta_j^*)$ is over-specified, i.e. fitted by at least two experts, then its estimation rate is of order $\mathcal{O}_P([\log(n)/n]^{\frac{1}{4}})$.

**(ii)** Secondly, for exact-specified parameters $\beta_{1j}^*, \eta_j^*$, where $\bar{k} + 1 \le j \le k_*$, the rates for estimating them are faster than those of their over-specified counterparts in Remark (i), standing at order $\mathcal{O}_P([\log(n)/n]^{\frac{1}{2}})$. By arguing as in equation (7), we deduce that the expert $h(\cdot, \eta_j^*)$ also enjoys the faster estimation rate of order $\mathcal{O}_P([\log(n)/n]^{\frac{1}{2}})$, which is parametric on the sample size $n$.

**(iii)** Putting the above two remarks together, we observe that when the expert functions are formulated as feed-forward networks with a widely used activation, namely ReLU and GELU, their estimation rates under the sigmoid gating MoE matches exactly those under the softmax gating MoE: see Theorem 3.2 in [26].

**Non-strongly identifiable experts.** The strong identifiability condition of Definition 1 is far from a technical requirement: it is in fact a crucial assumption. To illustrate its importance, we investigate the estimation rates of *ridge experts* of the form $\{h(x, (a_j^*, b_j^*)) = \varphi((a_j^*)^\top x + b_j^*), j \in [k_*]\}$, where $\varphi$ is a scalar function, that fail to satisfy the strong identifiability condition. For ridge experts, the strong identifiability will not be met in two cases: when at least one among over-specified parameters $a_j^*$ equals $0_d$, for any arbitrary activation function $\varphi$ (**Scenario I**) and when $\varphi$ is a polynomial of the form $\varphi(z) = z^p$, for $p \in \mathbb{N}$ (**Scenario II**).

**Scenario I.** Without loss of generality, we may assume that $a_1^* = 0_d$, which means that the first expert $\varphi((a_1^*)^\top x + b_1^*)$ becomes independent of the input $x$. This gives rise to an interaction among the gating parameter $\beta_1$ and the expert parameter $a$ via the partial differential equation (PDE)

$$\frac{\partial F}{\partial \beta_1}(x; \beta_{11}^*, \beta_{01}^*, a_1^*, b_1^*) = C_{b_1^*, \beta_{01}^*} \cdot \frac{\partial F}{\partial a}(x; \beta_{11}^*, \beta_{01}^*, a_1^*, b_1^*), \tag{8}$$

where we recall that $F(x; \beta_1, \beta_0, a, b) := \sigma(x; \beta_1, \beta_0)\varphi(a^\top x + b)$, and $C_{b_1^*, \beta_{01}^*}$ is some constant depending on $b_1^*$ and $\beta_{01}^*$. The above PDE can be verified by taking the derivatives of $F(x; \beta_1, \beta_0, a, b)$ w.r.t $\beta_1$ and $a$ and using the fact that $\beta_{11}^* = a_1^* = 0_d$. Notably, the PDE (8) accounts for the violation of the strong identifiability of the expert $\varphi((a_1^*)^\top x + b_1^*)$ under the Scenario I. To this end, we propose the following Voronoi loss to analyze the effects of such parameter interaction on the expert estimation rates in Theorem 3:

$$\mathcal{D}_{2,r}(G, G_*) := \sum_{j=1}^{\bar{k}} \Big| \sum_{i \in \mathcal{A}_j} \frac{1}{1 + \exp(-\beta_{0i})} - \frac{1}{1 + \exp(-\beta_{0j}^*)} \Big| + \sum_{j=1}^{\bar{k}} \sum_{i \in \mathcal{A}_j} \Big[ \|\Delta\beta_{1ij}\|^r + \|\Delta a_{ij}\|^r$$

$$+ |\Delta b_{ij}|^r \Big] + \sum_{j=\bar{k}+1}^{k_*} \sum_{i \in \mathcal{A}_j} \Big[ |\Delta\beta_{0ij}|^r + \|\Delta\beta_{1ij}\|^r + \|\Delta a_{ij}\|^r + |\Delta b_{ij}|^r \Big]. \tag{9}$$

The following result establishes a minimax lower bound on the estimation error of $G_*$ in the $\mathcal{D}_{2,r}$ loss.

**Theorem 3.** *Suppose that the experts take the form $\varphi(a^\top x + b)$. Then, under the Scenario I, we have*

$$\inf_{\widetilde{G}_n \in \mathcal{G}_k(\Theta)} \sup_{G \in \mathcal{G}_k(\Theta) \setminus \mathcal{G}_{k_*-1}(\Theta)} \mathbb{E}_{f_G}[\mathcal{D}_{2,r}(\widetilde{G}_n, G)] \gtrsim n^{-1/2},$$

*for any $r \geq 1$, where $\mathbb{E}_{f_G}$ indicates that the expectation taken w.r.t the product measure with $f_G^n$, and the infimum is over all estimators taking values in $\mathcal{G}_k(\Theta)$.*

See Appendix A.3 for the proof details. We highlight some important implications of Theorem 3.

**(i)** The estimation rates for the parameters $\beta_{1j}^*$, $a_j^*$ and $b_j^*$ are slower than $\mathcal{O}_P(n^{-1/2r})$, for any $r \geq 1$. This means that they are slower than any polynomial rates, and could be of order $\mathcal{O}_P(1/\log(n))$.

**(ii)** Using the same reasoning described after equation (7), we have

$$\sup_x |\varphi((\widehat{a}_i^n)^\top x + \widehat{b}_i^n) - \varphi((a_j^*)^\top x + b_j^*)| \leq L_2 \cdot (\|\widehat{a}_i^n - a_j^*\| + |\widehat{b}_i^n - b_j^*|), \tag{10}$$

where $L_2 > 0$ is a Lipschitz constant. As a consequence, the rates for estimating experts $\varphi((a_j^*)^\top x + b_j^*)$ are no better than those for estimating the parameters $a_j^*$ and $b_j^*$, and could also be as slow as $\mathcal{O}_P(1/\log(n))$. This result suggests that all the expert parameters $a_1^*, \ldots, a_{k_*}^*$ should be different from $0_d$ to hope for fast convergence rates. In other words, every expert of the form $\varphi(a^\top x + b)$ in the MoE model should vary with the input value.

**Scenario II.** Now, we consider the second scenario when $\varphi$ is a polynomial of the form $\varphi(z) = z^p$, for $p \in \mathbb{N}$. For simplicity, we will only focus on the setting of $p = 1$, i.e., $h(x, \eta_j^*) = (a_j^*)^\top x + b_j^*$; the case of $p > 1$ can be argued in a very similar fashion. The structure of the polynomial experts leads to a non-linear interactions among the parameters, expressed through the PDE

$$\frac{\partial^2 F}{\partial \beta_1 \partial b}(x; \beta_{1i}^*, a_i^*, b_i^*) = \frac{\partial^2 F}{\partial a \partial \beta_0}(x; \beta_{1i}^*, a_i^*, b_i^*), \tag{11}$$

where $F(x; \beta_1, \beta_0, a, b) := \sigma(x; \beta_1, \beta_0)(a^\top x + b)$. This interaction is the reason why the polynomial experts are not strongly identifiable. Ultimately, this dependence yields slow rates of convergence for expert estimation, just like in Scenario I, as revealed by the next result, which deploys the Voronoi loss $\mathcal{D}_{2,r}$ of equation (9).

**Theorem 4.** *Suppose that the experts take the form $(a^\top x + b)^p$, for some $p \in \mathbb{N}$. Then, under the Scenario II and for any $r \geq 1$,*

$$\inf_{\widetilde{G}_n \in \mathcal{G}_k(\Theta)} \sup_{G \in \mathcal{G}_k(\Theta) \setminus \mathcal{G}_{k_*-1}(\Theta)} \mathbb{E}_{f_G}[\mathcal{D}_{2,r}(\widetilde{G}_n, G)] \gtrsim n^{-1/2},$$

*where $\mathbb{E}_{f_G}$ indicates the expectation taken w.r.t the product measure with $f_G^n$.*

The proof of Theorem 4 is in Appendix A.4. To sum up, when either the experts are independent of the input or they take a polynomial form, their estimation rates could be as slow as $\mathcal{O}_P([\log(n)/n]^{\frac{1}{2}})$.

## 3.2 Regime 2 of Gating Parameters

Recall that under the Regime 2, at least one among the over-specified parameters $\beta^*_{1i}$ is different from $0_d$. As discussed in Section 2, the least squares regression estimator $f_{\widehat{G}_n}$ in this case converges to a regression function $f_{\overline{G}}$, where $\overline{G} \in \overline{\mathcal{G}}_k(\Theta) := \arg\min_{G \in \mathcal{G}_k(\Theta) \setminus \mathcal{G}_{k_*}(\Theta)} \|f_G - f_{G_*}\|_{L^2(\mu)}$. That is, the estimators of the parameters specifying $f_{\widehat{G}_n}$ converge to the parameters of $f_{\overline{G}}$. WLOG, we may assume that $\overline{G} := \sum_{i=1}^k \frac{1}{1+\exp(-\bar{\beta}_{0i})} \delta_{(\bar{\beta}_{1i}, \bar{\eta}_i)}$.

Similarly to Regime 1, we also provide in Definition 2 a *weak identifiability* condition to characterize those expert functions that will have fast estimation rates under the Regime 2. Weakly identifiable experts are required to satisfy only a subset of the conditions imposed to strongly identifiable experts.

**Definition 2** (Weak identifiability). *An expert function $x \mapsto h(x, \eta)$ is called weakly identifiable if it is twice differentiable w.r.t its parameter $\eta$ for $\mu$-almost all $x$ and, for any positive integer $\ell$ and any pair-wise distinct choices of parameters $\{(\beta_{0i}, \beta_{1i}, \eta_i)\}_{i=1}^\ell$, the functions in the class*

$$\left\{ x \mapsto \frac{\partial F}{\partial \beta_1^{\alpha_1} \partial \beta_0^{\alpha_2} \partial \eta^{\alpha_3}}(x, \beta_{1i}, \beta_{0i}, \eta_i) : i \in [\ell], (\alpha_1, \alpha_2, \alpha_3) \in \mathbb{N}^d \times \mathbb{N} \times \mathbb{N}^q, |\alpha_1| + \alpha_2 + |\alpha_3| = 1 \right\}$$

*are linearly independent, for $\mu$-almost all $x$.*

**Example.** Let us take an expert network $h(x, (a, b)) = \varphi(a^\top x + b)$ as an example. It can be checked that if $a \neq 0_d$ and the activation function $\varphi$ is a ReLU or GELU or a polynomial, then the expert $h(x, (a, b))$ is weakly identifiable. Conversely, if $a = 0_d$, i.e. the expert does not depend on the input, then the weak identifiability condition is not met, regardless of the choice of the activation.

We now describe the convergence rates for expert estimation under Regime 2. Since the analysis of input-independent experts can be done similarly to Theorem 3, we will focus only on weakly identifiable experts. As it turned out, the appropriate Voronoi loss function is given by

$$\mathcal{D}_3(G, \overline{G}) := \sum_{j=1}^k \sum_{i \in \mathcal{A}_j} \left[ |\beta_{0i} - \bar{\beta}_{0j}| + \|\beta_{1i} - \bar{\beta}_{1j}\| + \|\eta_i - \bar{\eta}_j\| \right]. \tag{12}$$

**Theorem 5.** *Let $h(x, \eta)$ be a weakly identifiable expert. Then, for any mixing measure $G \in \mathcal{G}_k(\Theta)$,*

$$\inf_{\overline{G} \in \overline{\mathcal{G}}_k(\Theta)} \|f_G - f_{\overline{G}}\|_{L^2(\mu)} / \mathcal{D}_3(G, \overline{G}) > 0.$$

*As a consequence, we obtain that $\inf_{\overline{G} \in \overline{\mathcal{G}}_k(\Theta)} \mathcal{D}_3(\widehat{G}_n, \overline{G}) = \mathcal{O}_P([\log(n)/n]^{\frac{1}{2}})$.*

The proof of Theorem 5 is in Appendix A.5. As a direct consequence of the bound and the definition of $\mathcal{D}_3$, the convergence rates of the estimators $\hat{\beta}_{0i}^n$, $\hat{\beta}_{1i}^n$ and $\hat{\eta}_i^n$ are of the same of parametric order $\mathcal{O}_P([\log(n)/n]^{\frac{1}{2}})$. Furthermore, by equation (7), we also establish that the convergence rate of the expert estimator $h(x, \hat{\eta}_i^n)$ is also of order $\mathcal{O}_P([\log(n)/n]^{\frac{1}{2}})$. Comparing these estimation rates with those established by [26] assuming instead softmax gating, there are two main observations:

**(i)** When the expert is a neural network with ReLU or GELU activation, the above parametric expert estimation rate obtained using the sigmoid gate is faster than the rate guranteed by the softmax gate, which could be of order $\mathcal{O}_P([\log(n)/n]^{\frac{1}{4}})$ (see [Theorem 3.2, [26]]).

**(ii)** When the activation is a polynomial, the gap between expert estimation rates when using the sigmoid gate versus when using the softmax gate becomes even more dramatic: $\mathcal{O}_P([\log(n)/n]^{\frac{1}{2}})$ compared to $\mathcal{O}_P(1/\log(n))$ (see Theorem 4.6 in [26]).

The above remarks highlight the considerable benefits of deploying sigmoid gating over softmax gating in MoE models: provably faster estimation rates not only for expert networks with popular activations like ReLU and GELU, but also for those with polynomial activation.

Table 2: True Parameters for Gating and Experts. The variance for the gating parameters is $\nu_g = 0.01/d$ and for the expert parameters is $\nu_e = 1/d$, where $d = 32$.

| Gating parameters | | Expert parameters | | |
|---|---|---|---|---|
| $\begin{cases} \beta_{1i}^* \sim \mathcal{N}(0_d, \nu_g I_d) & 1 \leq i \leq 7 \\ \beta_{1i}^* = 0_d & i = 8 \end{cases}$ | | $\beta_{0i}^* \sim \mathcal{N}(0, \nu_g)$ | $a_i^* \sim \mathcal{N}(0_d, \nu_e I_d)$ | $b_i^* \sim \mathcal{N}(0, \nu_e)$ |

## 4 Numerical Experiments

In this section, we perform some numerical experiments to empirically demonstrate our claim that the sigmoid gating is more sample efficient than the softmax gating.

From Table 1, it can be seen that the sigmoid gating shares the same expert estimation rates as the softmax gating under the Regime 1. However, the former gating outperforms the latter under the Regime 2, particularly for ReLU experts and polynomial experts. Therefore, we will consider those experts under the Regime 2 in our subsequent experiments.

**Data generation.** In particular, we generate the data by first sampling $X_i \sim \text{Uniform}([-1, 1]^d)$ for $i = 1, \ldots, n$. Then, we generate $Y_i$ according to the following model:

$$Y_i = g_{G_*}(X_i) + \epsilon_i, \quad i = 1, \ldots, n, \tag{13}$$

where the regression function $g_{G_*}(\cdot)$ take the form of a softmax gating MoE:

$$g_{G_*}(x) := \sum_{i=1}^{k_*} \text{Softmax}((\beta_{1i}^*)^\top x + \beta_{0i}^*) \cdot \phi\left((a_i^*)^\top x + b_i^*\right), \tag{14}$$

The input data dimension is $d = 32$. We employ $k_* = 8$ experts of the form $\phi(a^\top x + b)$, where $\phi$ is either the identity function or the ReLU function. The variance of Gaussian noise $\epsilon_i$ is $\nu = 0.01$.

**Experimental setup.** We summarize the choices of the ground-truth parameters $\beta_{0i}^*, \beta_{1i}^*, a_i^*$ and $b_i^*$ for $1 \leq i \leq 8$ in Table 2, which satisfies the condition of the Regime 2.

**Training procedure.** For each sample size $n$, spanning from $10^3$ to $10^5$, we perform 20 experiments. In every experiment, we employ $k = k_* + 1 = 9$ fitted experts, and the parameters initialization for the gating's and experts' parameters are adjusted to be near the true parameters, minimizing potential instabilities from the optimization process. Subsequently, we execute the stochastic gradient descent algorithm across 10 epochs, employing a learning rate of $\eta = 0.1$ to fit a model to the synthetic data. For each experiment, we calculate the Voronoi losses for every model and report the mean values for each sample size in Figure 1. Error bars representing two standard deviations are also shown.

**Results.** In Figure 1, when employing the ReLU experts, that is, $\phi$ is the ReLU function, the Voronoi loss for the sigmoid gating approaches zero at the rate of order $\mathcal{O}(n^{-0.51})$, which nearly matches our theoretical results in Theorem 5. Meanwhile, the loss for the softmax gating converges to zero at the slower rate $\mathcal{O}(n^{-0.24})$. On the other hand, when using the linear experts, that is, $\phi$ is the identity function, the vanishing rate of the Voronoi loss associated with the sigmoid gating is $\mathcal{O}(n^{-0.40})$, while that for the softmax gating is significantly slower, standing at $\mathcal{O}(n^{-0.11})$. These observations empirically shows that the sigmoid gating is more sample efficient than the softmax gating.

## 5 Discussion

In this paper, we carry out a convergence analysis of least squares expert estimation under MoE models with the sigmoid gating, which was found empirically to be robust to the representation collapse issue and to have a favorable performance in the MoE applications. We demonstrate that under both the gating regimes, the sigmoid gating requires a smaller sample size than the softmax gating to reach the same expert estimation error: that is, the sigmoid gating is more sample efficient than its softmax counterpart. Furthermore, we also verify that experts formulated as feed-forward networks with popular activation functions such as ReLU and GELU are more compatible with the sigmoid gating than other types of expert functions.

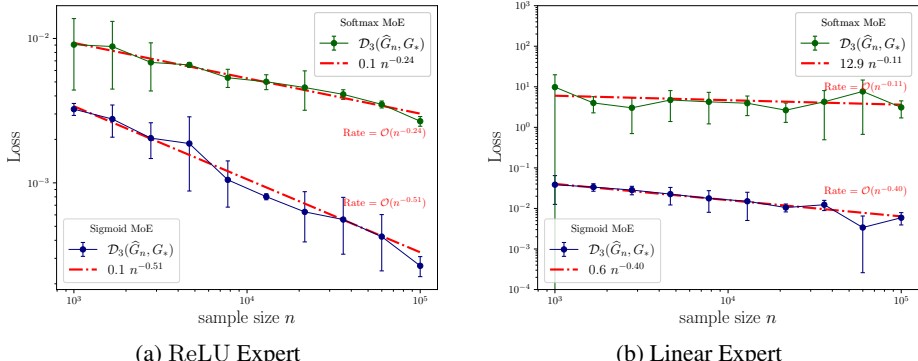

(a) ReLU Expert            (b) Linear Expert

Figure 1: Log-log scaled plots displaying the empirical averages of the Voronoi losses when using the sigmoid gating (blue line) versus when using the sofmax gating (green line) under the same data. The red dash-dotted lines illustrate the fitted lines for determining the empirical convergence rates.

**Practical implications.** Our theoretical findings support the following conclusions, which have considerable practical relevance.

**(I.1) The sigmoid gate is more sample efficient than the softmax gate for expert estimation.** It can be seen from Table 1 that while the expert estimation rates obtained when using the sigmoid gate match those attained when using the softmax gate under the Regime 1, the former are totally faster than the latter under the Regime 2. Notably, Regime 2 is closer to practice since the gating values under this regime hinge upon the input while those under the Regime 1 are input-independent. As a consequence, we can claim that the sigmoid gate is more sample efficient than the softmax gate from the perspective of the expert estimation problem.

**(I.2) The sigmoid gate is compatible with a broader class of experts than the softmax gate.** It follows from the expert characterization for Regime 1 (resp. Regime 2) in Definition 1 (resp. Definition 2) that formulating expert functions as feed-forward networks [36] with commonly used activation functions such as ReLU and GELU or even a polynomial activation will lead to faster expert estimation rates, and thus, require a smaller sample size to reach the same tolerance of estimating experts compared to the softmax gate. Thus, our theories indicate that the sigmoid gating is compatible with a broader class of experts than the softmax gating. This implication is particularly useful when people employ a mixture of fine-grained (shallow) expert networks [13].

**Limitations and future directions.** There are two main limitations of our current analysis:

1. We assume implicitly that the ground-truth parameters are independent of the sample size. Therefore, the expert estimation rates established in the paper are point-wise rather than the desirable uniform rates. A potential approach to cope with this problem is using the techniques for deriving the uniform parameter estimation rates in mixture models [6] and in MoE models [37]. However, those techniques are only valid for input-independent gating MoE, and we believe that further technical tools should be developed to adapt such framework to the sigmoid gating MoE. Thus, we leave it for future development.

2. The assumption that the true regression function belongs to the parametric class of MoE models under sigmoid gating is, of course, quite restrictive, and likely to be violated in real-world settings. This issue can be alleviated by assuming the data are sampled from a regression framework with an arbitrary regression function $g$, which is not necessarily formulated as a mixture of experts. Under that setting, the least squares estimator $\widehat{G}_n$ converges to $\widetilde{G} \in \arg\min_{G \in \mathcal{G}_k(\Theta)} \|f_G - g\|_{L^2(\mu)}$. Nevertheless, it requires the comprehensive knowledge of the universal approximation power of the sigmoid function, which has been slightly explored in [1], to determine the expert estimation rate. Therefore, we leave this potential direction for future work.

## Acknowledgements

NH acknowledges support from the NSF IFML 2019844 and the NSF AI Institute for Foundations of Machine Learning.

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

# Supplement to "Sigmoid Gating is More Sample Efficient than Softmax Gating in Mixture of Experts"

In this supplementary material, we present proofs for the main results of the paper in Appendix A, while we study the identifiability of the sigmoid gating mixture of experts (MoE) in Appendix B. Lastly, we provide additional numerical experiments to verify our theory in Appendix C.

## A    Proof of Main Results

In this appendix, we provide proofs for the theoretical results introduced in Section 2 and Section 3 of the paper.

### A.1    Proof of Theorem 1

To streamline the arguments for this proof, we need to define some necessary notations. First of all, we let $\mathcal{F}_k(\Theta) := \{f_G(x) : G \in \mathcal{G}_k(\Theta)\}$ stand for the set of all regression functions in $\mathcal{G}_k(\Theta)$. Next, for each $\delta > 0$, we define the $L^2(\mu)$-ball centered around the regression function $f_{G_*}(x)$ and intersected with the set $\mathcal{F}_k(\Theta)$ as

$$\mathcal{F}_k(\Theta, \delta) := \left\{ f \in \mathcal{F}_k(\Theta) : \|f - f_{G_*}\|_{L^2(\mu)} \leq \delta \right\}.$$

Furthermore, the size of the above ball is captured by the following integral as suggested in [35].

$$\mathcal{J}_B(\delta, \mathcal{F}_k(\Theta, \delta)) := \int_{\delta^2/2^{13}}^{\delta} H_B^{1/2}(t, \mathcal{F}_k(\Theta, t), \| \cdot \|_{L^2(\mu)}) \, \mathrm{d}t \vee \delta, \tag{15}$$

in which $H_B(t, \mathcal{F}_k(\Theta, t), \| \cdot \|_{L^2(\mu)})$ denotes the bracketing entropy [35] of $\mathcal{F}_k(\Theta, t)$ under the $L^2(\mu)$-norm, and $t \vee \delta := \max\{t, \delta\}$. By arguing in a similar fashion to Theorem 7.4 and Theorem 9.2 in [35] with adapted notations to this work, we achieve the following lemma:

**Lemma 1.** *Let $\Psi(\delta) \geq \mathcal{J}_B(\delta, \mathcal{F}_k(\Theta, \delta))$ such that $\Psi(\delta)/\delta^2$ is a non-increasing function of $\delta$. Then, for some universal constant $c$ and for some sequence $(\delta_n)$ that satisfies $\sqrt{n}\delta_n^2 \geq c\Psi(\delta_n)$, the following holds for any $\delta \geq \delta_n$:*

$$\mathbb{P}\Big(\|f_{\widehat{G}_n} - f_{G_*}\|_{L^2(\mu)} > \delta\Big) \leq c\exp\left(-\frac{n\delta^2}{c^2}\right).$$

**General Picture.** If we are able to demonstrate that the following bound holds for any $\varepsilon \in (0, 1/2]$:

$$H_B(\varepsilon, \mathcal{F}_k(\Theta), \|.\|_{L^2(\mu)}) \lesssim \log(1/\varepsilon), \tag{16}$$

then it follows that

$$\mathcal{J}_B(\delta, \mathcal{F}_k(\Theta, \delta)) = \int_{\delta^2/2^{13}}^{\delta} H_B^{1/2}(t, \mathcal{F}_k(\Theta, t), \| \cdot \|_{L^2(\mu)}) \, \mathrm{d}t \vee \delta$$

$$\lesssim \int_{\delta^2/2^{13}}^{\delta} \log(1/t) dt \vee \delta. \tag{17}$$

By choosing $\Psi(\delta) = \delta \cdot [\log(1/\delta)]^{1/2}$, then $\Psi(\delta)/\delta^2$ is a non-increasing function of $\delta$. Moreover, equation (17) suggests that $\Psi(\delta) \geq \mathcal{J}_B(\delta, \mathcal{F}_k(\Theta, \delta))$. Additionally, we set $\delta_n = \sqrt{\log(n)/n}$, and then get that $\sqrt{n}\delta_n^2 \geq c\Psi(\delta_n)$ for some universal constant $c$. According to Lemma 1, we achieve the desired conclusion of the theorem. As a consequence, it suffices to establish the bound (16).

**Proof of inequality** (16). Note that as the expert functions are bounded, we have that $|f_G(x)| \leq M$ for almost every $x$, where $M > 0$ is a bounded constant.

Next, let $\tau \leq \varepsilon$ and $\{\zeta_1, \ldots, \zeta_N\}$ be the $\tau$-cover under the $L^2(\mu)$-norm of the set $\mathcal{F}_k(\Theta)$, where $N := N(\tau, \mathcal{F}_k(\Theta), \| \cdot \|_{L^\infty})$ stands for the $\eta$-covering number of the metric space $(\mathcal{F}_k(\Theta), \| \cdot \|_{L^\infty})$. Now, we consider brackets of the form $[L_i(x), U_i(x)]$, where we define

$$L_i(x) := \max\{\zeta_i(x) - \tau, 0\}, \qquad U_i(x) := \max\{\zeta_i(x) + \tau, M\},$$

for all $i \in [N]$. Then, it can be verified that $\mathcal{F}_k(\Theta) \subset \cup_{i=1}^N [L_i(x), U_i(x)]$ and $U_i(x) - L_i(x) \leq \min\{2\tau, M\}$. Thus, we obtain that

$$\|U_i - L_i\|_{L^2(\mu)}^2 = \int (U_i - L_i)^2 \mathrm{d}\mu(x) \leq \int 4\tau^2 \mathrm{d}\mu(x) = 4\tau^2,$$

which indicates that $\|U_i - L_i\|_{L^2(\mu)} \leq 2\tau$. From the definition of the bracketing entropy, we arrive at

$$H_B(2\tau, \mathcal{F}_k(\Theta), \|\cdot\|_{L^2(\mu)}) \leq \log N = \log N(\tau, \mathcal{F}_k(\Theta), \|\cdot\|_{L^\infty}). \tag{18}$$

Consequently, it is sufficient to provide an upper bound for the covering number $N$. For that purpose, let us denote $\Delta := \{(\beta_0, \beta_1) \in \mathbb{R} \times \mathbb{R}^d : (\beta_0, \beta_1, \eta) \in \Theta\}$ and $\Omega := \{\eta \in \mathbb{R}^q : (\beta_0, \beta_1, \eta) \in \Theta\}$. Recall that $\Theta$ is a compact set, therefore, $\Delta$ and $\Omega$ are also compact. Then, there exist $\tau$-covers $\Delta_\tau$ and $\Omega_\tau$ for $\Delta$ and $\Omega$, respectively. Furthermore, it can be validated that

$$|\Delta_\tau| \leq \mathcal{O}_P(\tau^{-(d+1)k}), \quad |\Omega_\tau| \lesssim \mathcal{O}_P(\tau^{-qk}).$$

Given a mixing measure $G = \sum_{i=1}^k \exp(\beta_{0i})\delta_{(\beta_{1i}, \eta_i)} \in \mathcal{G}_k(\Theta)$, we take into account two other mixing measures defined as:

$$\widetilde{G} := \sum_{i=1}^k \frac{1}{1 + \exp(-\beta_{0i})}\delta_{(\beta_{1i}, \bar{\eta}_i)}, \qquad \check{G} := \sum_{i=1}^k \frac{1}{1 + \exp(-\check{\beta}_{0i})}\delta_{(\check{\beta}_{1i}, \check{\eta}_i)}.$$

Above, $\check{\eta}_i \in \Omega_\tau$ such that $\check{\eta}_i$ is the closest to $\eta_i$ in that set, while $(\check{\beta}_{0i}, \check{\beta}_{1i}) \in \Delta_\tau$ is the closest to $(\beta_{0i}, \beta_{1i})$ in that set. It follows from the above formulations that

$$\|f_G - f_{\widetilde{G}}\|_{L^\infty} = \sup_{x \in \mathcal{X}} \left| \sum_{i=1}^k \frac{1}{1 + \exp(-(\beta_{1i})^\top x - \beta_{0i})} \cdot [h(x, \eta_i) - h(x, \check{\eta}_i)] \right|$$

$$\leq \sum_{i=1}^k \sup_{x \in \mathcal{X}} \frac{1}{1 + \exp(-(\beta_{1i})^\top x - \beta_{0i})} \cdot |h(x, \eta_i) - h(x, \check{\eta}_i)|$$

$$\leq \sum_{i=1}^k \sup_{x \in \mathcal{X}} |h(x, \eta_i) - h(x, \check{\eta}_i)|$$

$$\leq \sum_{i=1}^k \sup_{x \in \mathcal{X}} L_1 \cdot \|\eta_i - \check{\eta}_i\|$$

$$\leq kL_1\tau \lesssim \tau.$$

Here, the second inequality occurs as the sigmoid weight is not larger than one and the third inequality follows from the fact that the expert function $h(x, \cdot)$ is a Lipschitz function with some Lipschitz constant $L_1 > 0$.

Subsequently, we have

$$\|f_{\widetilde{G}} - f_{\check{G}}\|_{L^\infty} = \sup_{x \in \mathcal{X}} \left| \sum_{i=1}^k \left[ \frac{1}{1 + \exp(-(\beta_{1i})^\top x - \beta_{0i})} - \frac{1}{1 + \exp(-(\check{\beta}_{1i})^\top x - \check{\beta}_{0i})} \right] \cdot h(x, \check{\eta}_i) \right|$$

$$\leq \sum_{i=1}^k \sup_{x \in \mathcal{X}} \left[ \frac{1}{1 + \exp(-(\beta_{1i})^\top x - \beta_{0i})} - \frac{1}{1 + \exp(-(\check{\beta}_{1i})^\top x - \check{\beta}_{0i})} \right] \cdot |h(x, \check{\eta}_i)|$$

$$\leq \sum_{i=1}^k \sup_{x \in \mathcal{X}} M' \left[ \frac{1}{1 + \exp(-(\beta_{1i})^\top x - \beta_{0i})} - \frac{1}{1 + \exp(-(\check{\beta}_{1i})^\top x - \check{\beta}_{0i})} \right]$$

$$\leq \sum_{i=1}^k \sup_{x \in \mathcal{X}} M' L_2(\|\beta_{1i} - \check{\beta}_{1i}\| \cdot \|x\| + |\beta_{0i} - \check{\beta}_{0i}|)$$

$$\leq kM'(\tau \cdot B + \tau) \lesssim \tau.$$

Above, the second inequality is due to the fact that the expert function is bounded, that is, $|h(x, \check{\eta}_i)| \leq M'$. The third one happens as the sigmoid function is a Lipschitz function with some Lipschitz constant $L_2 > 0$, while the fourth inequality occurs since the input space is bounded.

By the triangle inequality, we obtain

$$\|f_G - f_{\check{G}}\|_{L^\infty} \leq \|f_G - f_{\widetilde{G}}\|_{L^\infty} + \|f_{\widetilde{G}} - f_G\|_{L^\infty} \lesssim \tau.$$

From the definition of the covering number, it yields that

$$N(\tau, \mathcal{F}_k(\Theta), \|\cdot\|_{L^\infty} \leq |\Delta_\tau| \times |\Omega_\tau| \leq \mathcal{O}_P(n^{-(d+1)k}) \times \mathcal{O}(n^{-qk}) \leq \mathcal{O}(n^{-(d+1+q)k}). \quad (19)$$

Putting equations (18) and (19) together, we get that

$$H_B(2\tau, \mathcal{F}_k(\Theta), \|\cdot\|_{L^2(\mu)}) \lesssim \log(1/\tau).$$

Let $\tau = \varepsilon/2$, then we reach the desired bound

$$H_B(\varepsilon, \mathcal{F}_k(\Theta), \|.\|_{L^2(\mu)}) \lesssim \log(1/\varepsilon).$$

Hence, the proof is totally completed.

## A.2 Proof of Theorem 2

In this proof, we aim to establish the following inequality:

$$\inf_{G \in \mathcal{G}_k(\Theta)} \|f_G - f_{G_*}\|_{L^2(\mu)}/\mathcal{D}_1(G, G_*) > 0. \quad (20)$$

For that purpose, we divide the proof of the above inequality into local and global parts as below.

**Local part:** In this part, we focus only on demonstrating the following local inequality:

$$\lim_{\varepsilon \to 0} \inf_{G \in \mathcal{G}_k(\Theta):\mathcal{D}_1(G, G_*) \leq \varepsilon} \|f_G - f_{G_*}\|_{L^2(\mu)}/\mathcal{D}_1(G, G_*) > 0. \quad (21)$$

Assume by contrary that the above claim does not hold true, then there exists a sequence of mixing measures $G_n = \sum_{i=1}^{k_*} \delta_{(\beta_{0i}^n, \beta_{1i}^n, \eta_i^n)}$ in $\mathcal{G}_k(\Theta)$ such that $\mathcal{D}_{1n} := \mathcal{D}_1(G_n, G_*) \to 0$ and

$$\|f_{G_n} - f_{G_*}\|_{L^2(\mu)}/\mathcal{D}_{1n} \to 0, \quad (22)$$

as $n \to \infty$. Let us recall that

$$\mathcal{D}_{1n} := \sum_{j=1}^{\bar{k}} \Big| \sum_{i \in \mathcal{A}_j} \frac{1}{1 + \exp(-\beta_{0i}^n)} - \frac{1}{1 + \exp(-\beta_{0j}^*)} \Big| + \sum_{j=1}^{\bar{k}} \sum_{i \in \mathcal{A}_j} \Big[ \|\Delta\beta_{1ij}^n\|^2 + \|\Delta\eta_{ij}^n\|^2 \Big]$$

$$+ \sum_{j=\bar{k}+1}^{k_*} \sum_{i \in \mathcal{A}_j} \Big[ |\Delta\beta_{0ij}^n| + \|\Delta\beta_{1ij}^n\| + \|\Delta\eta_{ij}^n\| \Big],$$

where $\Delta\beta_{0ij}^n := \beta_{0i}^n - \beta_{0j}^*$, $\Delta\beta_{1ij}^n := \beta_{1i}^n - \beta_{1j}^*$ and $\Delta\eta_{ij}^n := \eta_i^n - \eta_j^*$.

Since $\mathcal{D}_{1n} \to 0$ as $n \to \infty$, we deduce that

- For $1 \leq j \leq \bar{k}$: $\sum_{i \in \mathcal{A}_j} \frac{1}{1+\exp(-\beta_{0i}^n)} \to \frac{1}{1+\exp(-\beta_{0j}^*)}$ and $(\beta_{1i}^n, \eta_i^n) \to (\beta_{1j}^*, \eta_j^*)$, for any $i \in \mathcal{A}_j$;

- For $\bar{k}+1 \leq j \leq k_*$: $(\beta_{0i}^n, \beta_{1i}^n, \eta_i^n) \to (\beta_{0j}^*, \beta_{1j}^*, \eta_j^*)$ for any $i \in \mathcal{A}_j$.

**Step 1 - Taylor expansion:** In this step, we decompose the term $f_{G_n}(x) - f_{G_*}(x)$ using Taylor expansion. Firstly, let us denote

$$f_{G_n}(x) - f_{G_*}(x) := \sum_{j=1}^{\bar{k}} \Big[ \sum_{i \in \mathcal{A}_j} \frac{h(x, \eta_i^n)}{1 + \exp(-(\beta_{1i}^n)^\top x - \beta_{0i}^n)} - \frac{h(x, \eta_j^*)}{1 + \exp(-\beta_{0j}^*)} \Big]$$

$$+ \sum_{j=\bar{k}+1}^{k_*} \Big[ \sum_{i \in \mathcal{A}_j} \frac{h(x, \eta_i^n)}{1 + \exp(-(\beta_{1i}^n)^\top x - \beta_{0i}^n)} - \frac{h(x, \eta_j^*)}{1 + \exp(-(\beta_{1j}^*)^\top x - \beta_{0j}^*)} \Big]$$

$$= \sum_{j=1}^{\bar{k}} \sum_{i \in \mathcal{A}_j} \Big[ \frac{h(x, \eta_i^n)}{1 + \exp(-(\beta_{1i}^n)^\top x - \beta_{0i}^n)} - \frac{h(x, \eta_j^*)}{1 + \exp(-\beta_{0i}^n)} \Big]$$

$$+ \sum_{j=1}^{\bar{k}} \Big[ \sum_{i \in \mathcal{A}_j} \frac{1}{1 + \exp(-\beta_{0i}^n)} - \frac{1}{1 + \exp(-\beta_{0j}^*)} \Big] \cdot h(x, \eta_j^*)$$

$$+ \sum_{j=\bar{k}+1}^{k_*} \Big[ \sum_{i \in \mathcal{A}_j} \frac{h(x, \eta_i^n)}{1 + \exp(-(\beta_{1i}^n)^\top x - \beta_{0i}^n)} - \frac{h(x, \eta_j^*)}{1 + \exp(-(\beta_{1j}^*)^\top x - \beta_{0j}^*)} \Big]$$

$$:= A_n(x) + B_n(x) + C_n(x). \tag{23}$$

Let us denote $\sigma(x, \beta_1, \beta_0) := \frac{1}{1 + \exp(-\beta_1^\top x - \beta_0)}$. Then, by means of the second-order Taylor expansion, we can decompose the term $A_n(x)$ defined above as

$$A_n(x) = \sum_{j=1}^{\bar{k}} \sum_{i \in \mathcal{A}_j} \Big[ \sigma(x, \beta_{1i}^n, \beta_{0i}^n) h(x, \eta_i^n) - \sigma(x, 0_d, \beta_{0i}^n) h(x, \eta_j^*) \Big]$$

$$= \sum_{j=1}^{\bar{k}} \sum_{i \in \mathcal{A}_j} \sum_{|\gamma|=1}^{2} \frac{1}{\gamma!} (\Delta\beta_{1ij}^n)^{\gamma_1} (\Delta\eta_{ij}^n)^{\gamma_2} \cdot \frac{\partial^{|\gamma_1|}\sigma}{\partial\beta_1^{\gamma_1}}(x, 0_d, \beta_{0i}^n) \frac{\partial^{|\gamma_2|}h}{\partial\eta^{\gamma_2}}(x, \eta_j^*) + R_1(x)$$

$$= \sum_{j=1}^{\bar{k}} \sum_{i \in \mathcal{A}_j} \sum_{|\gamma|=1}^{2} T_{j,i,\gamma_1,\gamma_2}^n \cdot \frac{\partial^{|\gamma_1|}\sigma}{\partial\beta_1^{\gamma_1}}(x, 0_d, \beta_{0i}^n) \frac{\partial^{|\gamma_2|}h}{\partial\eta^{\gamma_2}}(x, \eta_j^*) + R_1(x) \tag{24}$$

where $R_1(x)$ is a Taylor remainder such that $R_1(x)/\mathcal{D}_{1n} \to 0$ as $n \to \infty$, and we define

$$T_{j,i,\gamma_1,\gamma_2}^n := \frac{1}{\gamma!} (\Delta\beta_{1ij}^n)^{\gamma_1} (\Delta\eta_{ij}^n)^{\gamma_2},$$

for any $j \in [\bar{k}]$, $i \in \mathcal{A}_j$, $\gamma_1 \in \mathbb{N}^d$ and $\gamma_2 \in \mathbb{N}^q$ such that $1 \le |\gamma_1| + |\gamma_2| \le 2$.

Next, let us denote

$$W_j^n := \sum_{i \in \mathcal{A}_j} \frac{1}{1 + \exp(-\beta_{0i}^n)} - \frac{1}{1 + \exp(-\beta_{0j}^*)},$$

for any $j \in [\bar{k}]$, then the term $B_n(x)$ can be represented as

$$B_n(x) = \sum_{j=1}^{\bar{k}} W_j^n \cdot h(x, \eta_j^*). \tag{25}$$

Finally, recall that $|\mathcal{A}_j| = 1$ for any $\bar{k} + 1 \le j \le k_*$, we can decompose the term $C_n(x)$ by applying the first-order Taylor expansion as follows:

$$C_n(x) = \sum_{j=\bar{k}+1}^{k_*} \Big[ \sum_{i \in \mathcal{A}_j} \sigma(x, \beta_{1i}^n, \beta_{0i}^n) h(x, \eta_i^n) - \sigma(x, \beta_{1j}^*, \beta_{0j}^*) h(x, \eta_j^*) \Big]$$

$$= \sum_{j=\bar{k}+1}^{k_*} \sum_{|\alpha|=1} S_{j,\alpha_1,\alpha_2,\alpha_3}^n \cdot \frac{\partial^{|\alpha_1|+\alpha_2}\sigma}{\partial\beta_1^{\alpha_1}\partial\beta_0^{\alpha_2}}(x, \beta_{1j}^*, \beta_{0j}^*) \frac{\partial^{|\alpha_3|}h}{\partial\eta^{\alpha_3}}(x, \eta_j^*) + R_2(x), \tag{26}$$

where $R_2(x)$ is a Taylor remainder such that $R_2(x)/\mathcal{D}_{1n} \to 0$ as $n \to \infty$, and we define

$$S^n_{j,\alpha_1,\alpha_2,\alpha_3} := \sum_{i \in \mathcal{A}_j} \frac{1}{\alpha!} (\Delta\beta^n_{1ij})^{\alpha_1} (\Delta\beta^n_{0ij})^{\alpha_2} (\Delta\eta^n_{ij})^{\alpha_3},$$

for any $\bar{k}+1 \le j \le k_*$, $\alpha_1 \in \mathbb{N}^d$, $\alpha_2 \in \mathbb{N}$ and $\alpha_3 \in \mathbb{N}^q$ such that $|\alpha_1| + \alpha_2 + |\alpha_3| = 1$.

Combine the decomposition of the terms $A_n(x)$, $B_n(x)$ and $C_n(x)$ in equations (24), (25) and (26), we rewrite the difference $f_{G_n}(x) - f_{G_*}(x)$ as

$$f_{G_n}(x) - f_{G_*}(x) = \sum_{j=1}^{\bar{k}} \sum_{i \in \mathcal{A}_j} \sum_{|\gamma|=1}^{2} T^n_{j,i,\gamma_1,\gamma_2} \cdot \frac{\partial^{|\gamma_1|}\sigma}{\partial\beta_1^{\gamma_1}}(x, 0_d, \beta^n_{0i}) \frac{\partial^{|\gamma_2|}h}{\partial\eta^{\gamma_2}}(x, \eta^*_j) + \sum_{j=1}^{\bar{k}} W^n_j \cdot h(x, \eta^*_j)$$

$$+ \sum_{j=\bar{k}+1}^{k_*} \sum_{|\alpha|=1} S^n_{j,\alpha_1,\alpha_2,\alpha_3} \cdot \frac{\partial^{|\alpha_1|+\alpha_2}\sigma}{\partial\beta_1^{\alpha_1}\partial\beta_0^{\alpha_2}}(x, \beta^*_{1j}, \beta^*_{0j}) \frac{\partial^{|\alpha_3|}h}{\partial\eta^{\alpha_3}}(x, \eta^*_j) + R_1(x) + R_2(x). \tag{27}$$

**Step 2 - Non-vanishing coefficients:** In this step, we show that at least one among the ratios $T^n_{j,i,\gamma_1,\gamma_2}/\mathcal{D}_{1n}$, $W^n_j/\mathcal{D}_{1n}$ and $S^n_{j,\alpha_1,\alpha_2,\alpha_3}/\mathcal{D}_{1n}$ does not go to zero as $n$ tends to infinity. Indeed, assume by contrary that all of them converge to zero, i.e.

$$\frac{T^n_{j,i,\gamma_1,\gamma_2}}{\mathcal{D}_{1n}} \to 0, \quad \frac{W^n_j}{\mathcal{D}_{1n}} \to 0, \quad \frac{S^n_{j,\alpha_1,\alpha_2,\alpha_3}}{\mathcal{D}_{1n}} \to 0$$

as $n \to \infty$. Then, it follows that

- $\frac{1}{\mathcal{D}_{1n}} \sum_{j=1}^{\bar{k}} \left| \sum_{i \in \mathcal{A}_j} \frac{1}{1+\exp(-\beta^n_{0i})} - \frac{1}{1+\exp(-\beta^*_{0j})} \right| = \frac{1}{\mathcal{D}_{1n}} \cdot \sum_{j=1}^{\bar{k}} |W^n_j| \to 0$;

- $\frac{1}{\mathcal{D}_{1n}} \sum_{j=1}^{\bar{k}} \sum_{i \in \mathcal{A}_j} \|\Delta\beta^n_{1ij}\|^2 = \frac{1}{\mathcal{D}_{1n}} \cdot \sum_{j=1}^{\bar{k}} \sum_{i \in \mathcal{A}_j} \sum_{u=1}^{d} |T^n_{j,i,2e_{d,u},0_q}| \to 0$, where $e_{d,u}$ is a vector in $\mathbb{R}^d$ whose $u$-th entry is one while other entries are zero;

- $\frac{1}{\mathcal{D}_{1n}} \sum_{j=1}^{\bar{k}} \sum_{i \in \mathcal{A}_j} \|\Delta\eta^n_{ij}\|^2 = \frac{1}{\mathcal{D}_{1n}} \cdot \sum_{j=1}^{\bar{k}} \sum_{i \in \mathcal{A}_j} \sum_{v=1}^{q} |T^n_{j,i,0_d,2e_{q,v}}| \to 0$;

- $\frac{1}{\mathcal{D}_{1n}} \sum_{j=\bar{k}+1}^{k_*} \sum_{i \in \mathcal{A}_j} \|\Delta\beta^n_{1ij}\|_1 = \frac{1}{\mathcal{D}_{1n}} \cdot \sum_{j=\bar{k}+1}^{k_*} \sum_{u=1}^{d} |S^n_{j,e_{d,u},0,0_q}| \to 0$;

- $\frac{1}{\mathcal{D}_{1n}} \sum_{j=\bar{k}+1}^{k_*} \sum_{i \in \mathcal{A}_j} |\Delta\beta^n_{0ij}| = \frac{1}{\mathcal{D}_{1n}} \cdot \sum_{j=\bar{k}+1}^{k_*} |S^n_{j,0_d,1,0_q}| \to 0$;

- $\frac{1}{\mathcal{D}_{1n}} \sum_{j=\bar{k}+1}^{k_*} \sum_{i \in \mathcal{A}_j} \|\Delta\eta^n_{ij}\|_1 = \frac{1}{\mathcal{D}_{1n}} \cdot \sum_{j=\bar{k}+1}^{k_*} \sum_{v=1}^{q} |S^n_{j,0_d,0,e_{q,v}}| \to 0$.

By taking the summation of the above limits, we obtain that

$$\frac{1}{\mathcal{D}_{1n}} \left\{ \sum_{j=1}^{\bar{k}} \left| \sum_{i \in \mathcal{A}_j} \frac{1}{1+\exp(-\beta^n_{0i})} - \frac{1}{1+\exp(-\beta^*_{0j})} \right| + \sum_{j=1}^{\bar{k}} \sum_{i \in \mathcal{A}_j} \left[ \|\Delta\beta^n_{1ij}\|^2 + \|\Delta\eta^n_{ij}\|^2 \right] \right.$$

$$\left. + \sum_{j=\bar{k}+1}^{k_*} \sum_{i \in \mathcal{A}_j} \left[ |\Delta\beta^n_{0ij}| + \|\Delta\beta^n_{1ij}\|_1 + \|\Delta\eta^n_{ij}\|_1 \right] \right\} \to 0.$$

Due to the topological equivalence between the 1-norm and the 2-norm, we deduce that

$$1 = \frac{1}{\mathcal{D}_{1n}} \left\{ \sum_{j=1}^{\bar{k}} \left| \sum_{i \in \mathcal{A}_j} \frac{1}{1+\exp(-\beta^n_{0i})} - \frac{1}{1+\exp(-\beta^*_{0j})} \right| + \sum_{j=1}^{\bar{k}} \sum_{i \in \mathcal{A}_j} \left[ \|\Delta\beta^n_{1ij}\|^2 + \|\Delta\eta^n_{ij}\|^2 \right] \right.$$

$$\left. + \sum_{j=\bar{k}+1}^{k_*} \sum_{i \in \mathcal{A}_j} \left[ |\Delta\beta^n_{0ij}| + \|\Delta\beta^n_{1ij}\| + \|\Delta\eta^n_{ij}\| \right] \right\} \to 0,$$

which is a contradiction. As a consequence, at least one among the ratios $T^n_{j,i,\gamma_1,\gamma_2}/\mathcal{D}_{1n}$, $W^n_j/\mathcal{D}_{1n}$ and $S^n_{j,\alpha_1,\alpha_2,\alpha_3}/\mathcal{D}_{1n}$ must not approach zero as $n \to \infty$.

**Step 3 - Application of Fatou's lemma:** In this step, we use the Fatou's lemma to argue against the result in Step 2, and then, achieve the desired inequality in equation (21).

In particular, let us denote by $m_n$ the maximum of the absolute values of $T^n_{j,i,\gamma_1,\gamma_2}/\mathcal{D}_{1n}$, $W^n_j/\mathcal{D}_{1n}$ and $S^n_{j,\alpha_1,\alpha_2,\alpha_3}/\mathcal{D}_{1n}$. Since at least one among those ratios must not approach zero as $n \to \infty$, we get that $1/m_n \not\to \infty$ as $n \to \infty$.

Recall from the hypothesis in equation (22) that $\|f_{G_n} - f_{G_*}\|_{L^2(\mu)}/\mathcal{D}_{1n} \to 0$ as $n \to \infty$, which indicates that $\|f_{G_n} - f_{G_*}\|_{L^1(\mu)}/\mathcal{D}_{1n} \to 0$ due to the equivalence between $L^1(\mu)$-norm and $L^2(\mu)$-norm. By means of the Fatou's lemma, we have

$$0 = \lim_{n\to\infty} \frac{\|f_{G_n} - f_{G_*}\|_{L^1(\mu)}}{m_n \mathcal{D}_{1n}} \geq \int \liminf_{n\to\infty} \frac{|f_{G_n}(x) - f_{G_*}(x)|}{m_n \mathcal{D}_{1n}} d\mu(x) \geq 0.$$

This result implies that $[f_{G_n}(x) - f_{G_*}(x)]/[m_n \mathcal{D}_{1n}] \to 0$ for almost every $x$.

Let us denote

$$T^n_{j,i,\gamma_1,\gamma_2}/m_n \mathcal{D}_{1n} \to t_{j,i,\gamma_1,\gamma_2},$$
$$W^n_j/m_n \mathcal{D}_{1n} \to w_j,$$
$$S^n_{j,\alpha_1,\alpha_2,\alpha_3}/m_n \mathcal{D}_{1n} \to s_{j,\alpha_1,\alpha_2,\alpha_3},$$

with a note that at least one among the limits $t_{j,i,\gamma_1,\gamma_2}$, $w_j$ and $s_{j,\alpha_1,\alpha_2,\alpha_3}$ is non-zero. Then, from the decomposition in equation (27), we deduce that

$$\sum_{j=1}^{\bar{k}} \sum_{i\in\mathcal{A}_j} \sum_{|\gamma|=1}^{2} t_{j,i,\gamma_1,\gamma_2} \cdot \frac{\partial^{|\gamma_1|}\sigma}{\partial\beta_1^{\gamma_1}}(x, 0_d, \bar{\beta}_{0i}) \frac{\partial^{|\gamma_2|}h}{\partial\eta^{\gamma_2}}(x, \eta_j^*) + \sum_{j=1}^{\bar{k}} w_j \cdot h(x, \eta_j^*)$$
$$+ \sum_{j=\bar{k}+1}^{k_*} \sum_{|\alpha|=1} s_{j,\alpha_1,\alpha_2,\alpha_3} \cdot \frac{\partial^{|\alpha_1|+\alpha_2}\sigma}{\partial\beta_1^{\alpha_1}\partial\beta_0^{\alpha_2}}(x, \beta_{1j}^*, \beta_{0j}^*) \frac{\partial^{|\alpha_3|}h}{\partial\eta^{\alpha_3}}(x, \eta_j^*) = 0,$$

for almost every $x$. Note that the expert function $h(\cdot, \eta)$ is strongly identifiable, then the above equation implies that $t_{j,\gamma_1,\gamma_2} = w_j = s_{j,\alpha_1,\alpha_2,\alpha_3} = 0$, for any $j \in [k_*]$, $\alpha_1, \gamma_1 \in \mathbb{N}^d$, $\alpha_2 \in \mathbb{N}$ and $\gamma_2, \alpha_3 \in \mathbb{N}^q$ such that $1 \leq |\gamma_1| + |\gamma_2| \leq 2$ and $|\alpha_1| + \alpha_2 + |\alpha_3| = 1$. This contradicts the fact that at least one among the limits $s_{i,\alpha_1,\alpha_2,\alpha_3}$ is different from zero.

Hence, we obtain the local inequality in equation (21). Consequently, there exists some $\varepsilon' > 0$ such that

$$\inf_{G\in\mathcal{G}_k(\Theta):\mathcal{D}_1(G,G_*)\leq\varepsilon'} \|f_G - f_{G_*}\|_{L^2(\mu)}/\mathcal{D}_1(G, G_*) > 0.$$

**Global part:** Given the above result, it suffices to demonstrate that

$$\inf_{G\in\mathcal{G}_k(\Theta):\mathcal{D}_1(G,G_*)>\varepsilon'} \|f_G - f_{G_*}\|_{L^2(\mu)}/\mathcal{D}_1(G, G_*) > 0. \tag{28}$$

Assume by contrary that the inequality (28) does not hold true, then we can find a sequence of mixing measures $G'_n \in \mathcal{G}_k(\Theta)$ such that $\mathcal{D}_1(G'_n, G_*) > \varepsilon'$ and

$$\lim_{n\to\infty} \frac{\|f_{G'_n} - f_{G_*}\|_{L^2(\mu)}}{\mathcal{D}_1(G'_n, G_*)} = 0,$$

which indicates that $\|f_{G'_n} - f_{G_*}\|_{L^2(\mu)} \to 0$ as $n \to \infty$. Recall that $\Theta$ is a compact set, therefore, we can replace the sequence $G'_n$ by one of its subsequences that converges to a mixing measure $G' \in \mathcal{G}_k(\Omega)$. Since $\mathcal{D}_1(G'_n, G_*) > \varepsilon'$, we deduce that $\mathcal{D}_1(G', G_*) > \varepsilon'$.

Next, by invoking the Fatou's lemma, we have that

$$0 = \lim_{n\to\infty} \|f_{G'_n} - f_{G_*}\|_{L^2(\mu)}^2 \geq \int \liminf_{n\to\infty} \left|f_{G'_n}(x) - f_{G_*}(x)\right|^2 d\mu(x).$$

Thus, we get that $f_{G'}(x) = f_{G_*}(x)$ for almost every $x$. From Proposition 1, we deduce that $G' \equiv G_*$. Consequently, it follows that $\mathcal{D}_1(G', G_*) = 0$, contradicting the fact that $\mathcal{D}_1(G', G_*) > \varepsilon' > 0$.

Hence, the proof is completed.

### A.3 Proof of Theorem 3

**Lemma 2.** *If the following holds for any $r \geq 1$:*

$$\lim_{\varepsilon \to 0} \inf_{G \in \mathcal{G}_k(\Theta): \mathcal{D}_{2,r}(G, G_*) \leq \varepsilon} \frac{\|f_G - f_{G_*}\|_{L^2(\mu)}}{\mathcal{D}_{2,r}(G, G_*)} = 0, \tag{29}$$

*then we obtain that*

$$\inf_{\widetilde{G}_n \in \mathcal{G}_k(\Theta)} \sup_{G \in \mathcal{G}_k(\Theta) \setminus \mathcal{G}_{k_*-1}(\Theta)} \mathbb{E}_{f_G}[\mathcal{D}_{2,r}(\widetilde{G}_n, G)] \gtrsim n^{-1/2}, \tag{30}$$

*Proof of Lemma 2.* Indeed, from the Gaussian assumption on the noise variables $\epsilon_i$, we obtain that $Y_i | X_i \sim \mathcal{N}(f_{G_*}(X_i), \sigma^2)$ for all $i \in [n]$. Next, the assumption in equation (29) indicates for sufficiently small $\varepsilon > 0$ and a fixed constant $C_1 > 0$ which we will choose later, we can find a mixing measure $G'_* \in \mathcal{G}_k(\Theta)$ such that $\mathcal{D}_{2,r}(G'_*, G_*) = 2\varepsilon$ and $\|f_{G'_*} - f_{G_*}\|_{L^2(\mu)} \leq C_1 \varepsilon$. From Le Cam's lemma [39], as the Voronoi loss function $\mathcal{D}_{2,r}$ satisfies the weak triangle inequality, we obtain that

$$\inf_{\widetilde{G}_n \in \mathcal{G}_k(\Theta)} \sup_{G \in \mathcal{G}_k(\Theta) \setminus \mathcal{G}_{k_*-1}(\Theta)} \mathbb{E}_{f_G}[\mathcal{D}_{2,r}(\widetilde{G}_n, G)]$$

$$\gtrsim \frac{\mathcal{D}_{2,r}(G'_*, G_*)}{8} \exp(-n \mathbb{E}_{X \sim \mu}[\mathrm{KL}(\mathcal{N}(f_{G'_*}(X), \sigma^2), \mathcal{N}(f_{G_*}(X), \sigma^2))])$$

$$\gtrsim \varepsilon \cdot \exp(-n \|f_{G'_*} - f_{G_*}\|_{L^2(\mu)}^2),$$

$$\gtrsim \varepsilon \cdot \exp(-C_1 n \varepsilon^2), \tag{31}$$

where the second inequality is due to the fact that

$$\mathrm{KL}(\mathcal{N}(f_{G'_*}(X), \sigma^2), \mathcal{N}(f_{G_*}(X), \sigma^2)) = \frac{(f_{G'_*}(X) - f_{G_*}(X))^2}{2\sigma^2}.$$

By choosing $\varepsilon = n^{-1/2}$, we obtain that $\varepsilon \cdot \exp(-C_1 n \varepsilon^2) = n^{-1/2} \exp(-C_1)$. As a consequence, we achieve the desired minimax lower bound in equation (30). $\square$

**Main proof.** Following from the result of Lemma 2, it is sufficient to show that the following limit holds true for any $r \geq 1$:

$$\lim_{\varepsilon \to 0} \inf_{G \in \mathcal{G}_k(\Theta): \mathcal{D}_{2,r}(G, G_*) \leq \varepsilon} \frac{\|f_G - f_{G_*}\|_{L^2(\mu)}}{\mathcal{D}_{2,r}(G, G_*)} = 0. \tag{32}$$

To this end, we need to construct a sequence of mixing measures $(G_n)$ that satisfies $\mathcal{D}_{2,r}(G_n, G_*) \to 0$ and

$$\frac{\|f_{G_n} - f_{G_*}\|_{L^2(\mu)}}{\mathcal{D}_{2,r}(G_n, G_*)} \to 0,$$

as $n \to \infty$. Recall that under the Case 2, at least one among parameters $a_1^*, \ldots, a_{k_*}^*$ is equal to $0_d$. Without loss of generality, we may assume that $a_1^* = 0_d$. Next, let us take into account the sequence $(G_n)$ with $k_* + 1$ atoms in which

- $\exp(-\beta_{01}^n) = \exp(-\beta_{02}^n) = 1 + 2\exp(-\beta_{01}^*)$ and $\beta_{0i}^n = \beta_{0(i-1)}^*$ for any $3 \leq i \leq k_* + 1$;
- $\beta_{11}^n = \beta_{12}^n = \beta_{11}^* = 0_d$ and $\beta_{1i}^n = \beta_{1(i-1)}^*$ for any $3 \leq i \leq k_* + 1$;
- $a_1^n = a_2^n = a_1^* = 0_d$ and $a_i^n = a_{i-1}^*$ for any $3 \leq i \leq k_* + 1$;
- $b_1^n = b_1^* + \frac{c}{n}$, $b_2^n = b_1^* + \frac{2c}{n}$ and $b_i^n = b_{i-1}^*$ for any $3 \leq i \leq k_* + 1$,

where $c \in \mathbb{R}$ will be chosen later. Consequently, we get that

$$\mathcal{D}_{2,r}(G_n, G_*) = \frac{c^r}{n^r} + \frac{(2c)^r}{n^r} = \mathcal{O}(n^{-r}).$$

Next, we demonstrate that $\|f_{G_n} - f_{G_*}\|_{L^2(\mu)}/\mathcal{D}_{2,r}(G_n, G_*) \to 0$. In particular, following from the decomposition of $f_{G_n}(x) - f_{G_*}(x)$ in equation (23) and the above parameter setting, we have that

$$f_{G_n}(x) - f_{G_*}(x) = \sum_{i=1}^{2} \left[ \frac{\varphi((a_i^n)^\top x + b_i^n)}{1 + \exp(-(\beta_{1i}^n)^\top x - \beta_{0i}^n)} - \frac{\varphi((a_1^*)^\top x + b_1^*)}{1 + \exp(-\beta_{0i}^n)} \right]$$

$$+ \left[ \sum_{i=1}^{2} \frac{1}{1 + \exp(-\beta_{0i}^n)} - \frac{1}{1 + \exp(-\beta_{01}^*)} \right] \cdot \varphi((a_1^*)^\top x + b_1^*)$$

$$+ \sum_{i=3}^{k_*+1} \left[ \frac{\varphi((a_i^n)^\top x + b_i^n)}{1 + \exp(-(\beta_{1i}^n)^\top x - \beta_{0i}^n)} - \frac{\varphi((a_{i-1}^*)^\top x + b_{i-1}^*)}{1 + \exp(-(\beta_{1(i-1)}^*)^\top x - \beta_{0(i-1)}^*)} \right]$$

$$= \sum_{i=1}^{2} \left[ \frac{\varphi(b_i^n)}{1 + \exp(-\beta_{0i}^n)} - \frac{\varphi(b_1^*)}{1 + \exp(-\beta_{0i}^n)} \right]$$

$$= \frac{1}{2 + 2\exp(-\beta_{01}^*)} \cdot \sum_{i=1}^{2} [\varphi(b_i^n) - \varphi(b_1^*)].$$

Thus, we deduce that

$$[2 + 2\exp(-\beta_{01}^*)] \cdot [f_{G_n}(x) - f_{G_*}(x)] = \sum_{i=1}^{2} [\varphi(b_i^n) - \varphi(b_1^*)].$$

**When $r$ is an odd natural number:** By applying the Taylor expansion up to order $r$-th, we get that

$$[2 + 2\exp(-\beta_{01}^*)] \cdot [f_{G_n}(x) - f_{G_*}(x)] = \sum_{i=1}^{2} \sum_{\alpha=1}^{r} \frac{(b_i^n - b_1^*)^\alpha}{\alpha!} \varphi^{(\alpha)}(b_1^*) + R_1(x)$$

$$= \sum_{\alpha=1}^{r} \frac{(1 + 2^\alpha)c^\alpha}{\alpha! n^\alpha} \cdot \varphi^{(\alpha)}(b_1^*) + R_1(x),$$

where $R_1(x)$ is the Taylor remainder such that $R_1(x)/\mathcal{D}_{2,r}(G_n, G_*) \to 0$ as $n \to \infty$.

Note that $\left[ \sum_{\alpha=1}^{r} \frac{(1+2^\alpha)\varphi^{(\alpha)}(b_1^*)}{\alpha! n^\alpha} \cdot c^\alpha \right]$ is an odd-order polynomial of $c$. Thus, we can choose $c$ as a root of this polynomial, which leads to the fact that

$$f_{G_n}(x) - f_{G_*}(x) = \frac{R_1(x)}{2 + 2\exp(-\beta_{01}^*)}.$$

From the above results, we deduce that $[f_{G_n}(x) - f_{G_*}(x)]/\mathcal{D}_{2,r}(G_n, G_*) \to 0$ for almost every $x$. As a consequence, we achieve that $\|f_{G_n} - f_{G_*}\|_{L^2(\mu)}/\mathcal{D}_{2,r}(G_n, G_*) \to 0$ as $n \to \infty$.

**When $r$ is an even natural number:** By means of the Taylor expansion of order $(r+1)$-th, we have

$$[2 + 2\exp(-\beta_{01}^*)] \cdot [f_{G_n}(x) - f_{G_*}(x)] = \sum_{\alpha=1}^{r+1} \frac{(1 + 2^\alpha)c^\alpha}{\alpha! n^\alpha} \cdot \varphi^{(\alpha)}(b_1^*) + R_2(x),$$

where $R_2(x)$ is a Taylor remainder such that $R_2(x)/\mathcal{D}_{2,r}(G_n, G_*) \to 0$.

Since $\left[ \sum_{\alpha=1}^{r+1} \frac{(1+2^\alpha)\sigma^{(\alpha)}(b_1^*)}{\alpha! n^\alpha} \cdot c^\alpha \right]$ is an odd-degree polynomial of variable $c$, we can argue in a similar fashion to the scenario when $r$ is odd to obtain that $\|f_{G_n} - f_{G_*}\|_{L^2(\mu)}/\mathcal{D}_{2,r}(G_n, G_*) \to 0$.

Combine results from the above two scenarios of $r$, we obtain the claim in equation (32).

### A.4 Proof of Theorem 4

Following from the result of Lemma 2, it is sufficient to show that the following limit holds true for any $r \geq 1$:

$$\lim_{\varepsilon \to 0} \inf_{G \in \mathcal{G}_k(\Theta): \mathcal{D}_{2,r}(G,G_*) \leq \varepsilon} \frac{\|f_G - f_{G_*}\|_{L^2(\mu)}}{\mathcal{D}_{2,r}(G, G_*)} = 0. \tag{33}$$

To this end, we need to construct a sequence of mixing measures $(G_n)$ that satisfies $\mathcal{D}_{2,r}(G_n, G_*) \to 0$ and

$$\frac{\|f_{G_n} - f_{G_*}\|_{L^2(\mu)}}{\mathcal{D}_{2,r}(G_n, G_*)} \to 0,$$

as $n \to \infty$. Recall that under the Case 2, at least one among parameters $a_1^*, \ldots, a_{k_*}^*$ is equal to $0_d$. Without loss of generality, we may assume that $a_1^* = 0_d$. Next, let us take into account the sequence $(G_n)$ with $k_* + 1$ atoms in which

- $\beta_{01}^n = \beta_{02}^n$ such that

$$\sum_{i=1}^{2} \frac{1}{1 + \exp(-\beta_{0i}^n)} = \frac{1}{1 + \exp(-\beta_{01}^*)} + \frac{1}{n^{r+1}},$$

  and $\beta_{0i}^n = \beta_{0(i-1)}^*$ for any $3 \le i \le k_* + 1$;
- $\beta_{11}^n = \beta_{12}^n = \beta_{11}^* = 0_d$ and $\beta_{1i}^n = \beta_{1(i-1)}^*$ for any $3 \le i \le k_* + 1$;
- $a_1^n = a_2^n = a_1^*$ and $a_i^n = a_{i-1}^*$ for any $3 \le i \le k_* + 1$;
- $b_1^n = b_1^* + \frac{1}{n}$, $b_2^n = b_1^* - \frac{1}{n}$ and $b_i^n = b_{i-1}^*$ for any $3 \le i \le k_* + 1$,

Consequently, we get that

$$\mathcal{D}_{2,r}(G_n, G_*) = \frac{1}{n^{r+1}} + \frac{2}{n^r} = \mathcal{O}(n^{-r}).$$

Next, we demonstrate that $\|f_{G_n} - f_{G_*}\|_{L^2(\mu)} / \mathcal{D}_{2,r}(G_n, G_*) \to 0$. In particular, following from the decomposition of $f_{G_n}(x) - f_{G_*}(x)$ in equation (23) and the above parameter setting, we have that

$$
\begin{aligned}
f_{G_n}(x) - f_{G_*}(x) &= \sum_{i=1}^{2} \left[ \frac{(a_i^n)^\top x + b_i^n}{1 + \exp(-(\beta_{1i}^n)^\top x - \beta_{0i}^n)} - \frac{(a_1^*)^\top x + b_1^*}{1 + \exp(-\beta_{0i}^n)} \right] \\
&+ \left[ \sum_{i=1}^{2} \frac{1}{1 + \exp(-\beta_{0i}^n)} - \frac{1}{1 + \exp(-\beta_{01}^*)} \right] \cdot [(a_1^*)^\top x + b_1^*] \\
&+ \sum_{i=3}^{k_*+1} \left[ \frac{(a_i^n)^\top x + b_i^n}{1 + \exp(-(\beta_{1i}^n)^\top x - \beta_{0i}^n)} - \frac{(a_{i-1}^*)^\top x + b_{i-1}^*}{1 + \exp(-(\beta_{1(i-1)}^*)^\top x - \beta_{0(i-1)}^*)} \right] \\
&= \sum_{i=1}^{2} \left[ \frac{(a_1^*)^\top x + b_i^n}{1 + \exp(-\beta_{0i}^n)} - \frac{(a_1^*)^\top x + b_1^*}{1 + \exp(-\beta_{0i}^n)} \right] + \frac{1}{n^{r+1}} \cdot [(a_1^*)^\top x + b_1^*] \\
&= \frac{1}{1 + \exp(-\beta_{01}^n)} \cdot \sum_{i=1}^{2} [b_i^n - b_1^*] + \frac{1}{n^{r+1}} \cdot [(a_1^*)^\top x + b_1^*] \\
&= \frac{1}{n^{r+1}} \cdot [(a_1^*)^\top x + b_1^*].
\end{aligned}
$$

From the above result, we deduce that $[f_{G_n}(x) - f_{G_*}(x)] / \mathcal{D}_{2,r}(G_n, G_*) \to 0$ for almost every $x$. As a consequence, we achieve that $\|f_{G_n} - f_{G_*}\|_{L^2(\mu)} / \mathcal{D}_{2,r}(G_n, G_*) \to 0$ as $n \to \infty$. Hence, we obtain the claim in equation (33).

### A.5 Proof of Theorem 5

Following from the result of Corollary 1, it suffices to establish the following inequality:

$$\inf_{G \in \mathcal{G}_k(\Theta)} \|f_G - f_{\overline{G}}\|_{L^2(\mu)} / \mathcal{D}_3(G, \overline{G}) > 0, \tag{34}$$

for any mixing measure $\overline{G} \in \overline{\mathcal{G}}_k(\Theta)$. For that purpose, we divide the proof of the above inequality into local and global parts as in Appendix A.2. However, since the arguments for the global part remain the same (up to some changes of notations) for the over-specified setting, they will be omitted.

Therefore, given an arbitrary mixing measure $\overline{G} := \sum_{i=1}^{k} \frac{1}{1+\exp(-\bar{\beta}_{0i})} \delta_{(\bar{\beta}_{1i}, \bar{\eta}_i)} \in \overline{\mathcal{G}}_k(\Theta)$, we focus only on demonstrating that

$$\lim_{\varepsilon \to 0} \inf_{G \in \mathcal{G}_k(\Theta): \mathcal{D}_3(G,\overline{G}) \leq \varepsilon} \|f_G - f_{\overline{G}}\|_{L^2(\mu)}/\mathcal{D}_3(G,\overline{G}) > 0. \tag{35}$$

Assume by contrary that the above inequality does not hold true, then there exists a sequence of mixing measures $G_n = \sum_{i=1}^{k} \frac{1}{1+\exp(-\beta_{0i}^n)} \delta_{(\beta_{1i}^n, \eta_i^n)}$ in $\mathcal{G}_k(\Theta)$ such that $\mathcal{D}_{3n} := \mathcal{D}_3(G_n,\overline{G}) \to 0$ and

$$\|f_{G_n} - f_{\overline{G}}\|_{L^2(\mu)}/\mathcal{D}_{3n} \to 0, \tag{36}$$

as $n \to \infty$. Let us denote by $\mathcal{A}_i^n := \mathcal{A}_i(G_n)$ a Voronoi cell of $G_n$ generated by the $j$-th components of $\overline{G}$. Since our arguments are asymptotic, we may assume that those Voronoi cells do not depend on the sample size, i.e. $\mathcal{A}_j = \mathcal{A}_i^n$.

Additionally, since $G_n$ and $\overline{G}$ share the same number of atoms $k$ under the Regime 2 and $\mathcal{D}_{3n} \to 0$, the Voronoi cell $\mathcal{A}_i$ has only one element for any $i \in [k]$. WLOG, we assume that $\mathcal{A}_i = \{i\}$ for all $i \in [k]$, which follows that $(\beta_{0i}^n, \beta_{1i}^n, \eta_i^n) \to (\bar{\beta}_{0i}, \bar{\beta}_{1i}, \bar{\eta}_i)$ as $n \to \infty$ for any $i \in [k]$.

Thus, the Voronoi loss $\mathcal{D}_{3n}$ can be represented as

$$\mathcal{D}_{3n} := \sum_{i=1}^{k} \left[ |\Delta\bar{\beta}_{0i}^n| + \|\Delta\bar{\beta}_{1i}^n\| + \|\Delta\bar{\eta}_i^n\| \right], \tag{37}$$

where we denote $\Delta\bar{\beta}_{0i}^n := \beta_{0i}^n - \bar{\beta}_{0i}$, $\Delta\bar{\beta}_{1i}^n := \beta_{1i}^n - \bar{\beta}_{1i}$ and $\Delta\bar{\eta}_i^n := \eta_i^n - \bar{\eta}_i$.

Now, we divide the proof of local part into three steps as follows:

**Step 1 - Taylor expansion:** In this step, we decompose the term $f_{G_n}(x) - f_{\overline{G}}(x)$ using Taylor expansion. Let us denote $\sigma(x, \beta_1, \beta_0) := \frac{1}{1+\exp(-\beta_1^\top x - \beta_0)}$, then we have

$$f_{G_n}(x) - f_{\overline{G}}(x) := \sum_{i=1}^{k} \left[ \frac{h(x, \eta_i^n)}{1+\exp(-(\beta_{1i}^n)^\top x - \beta_{0i}^n)} - \frac{h(x, \bar{\eta}_i)}{1+\exp(-(\bar{\beta}_{1i})^\top x - \bar{\beta}_{0i})} \right]$$

$$= \sum_{i=1}^{k} \sum_{|\alpha|=1} \frac{1}{\alpha!} (\Delta\bar{\beta}_{1i}^n)^{\alpha_1} (\Delta\bar{\beta}_{0i}^n)^{\alpha_2} (\Delta\bar{\eta}_i^n)^{\alpha_3} \cdot \frac{\partial^{|\alpha_1|+\alpha_2}\sigma}{\partial\beta_1^{\alpha_1}\partial\beta_0^{\alpha_2}}(x, \bar{\beta}_{1i}, \bar{\beta}_{0i}) \frac{\partial^{|\alpha_3|}h}{\partial\eta^{\alpha_3}}(x, \bar{\eta}_i) + R_1(x)$$

$$= \sum_{i=1}^{k} \sum_{|\alpha|=1} S_{i,\alpha_1,\alpha_2,\alpha_3}^n \cdot \frac{\partial^{|\alpha_1|+\alpha_2}\sigma}{\partial\beta_1^{\alpha_1}\partial\beta_0^{\alpha_2}}(x, \bar{\beta}_{1i}, \bar{\beta}_{0i}) \frac{\partial^{|\alpha_3|}h}{\partial\eta^{\alpha_3}}(x, \bar{\eta}_i) + R_1(x), \tag{38}$$

where $R_1(x)$ is a Taylor remainder such that $R_1(x)/\mathcal{D}_{3n} \to 0$ as $n \to \infty$, and

$$S_{i,\alpha_1,\alpha_2,\alpha_3}^n := \sum_{|\alpha|=1} \frac{1}{\alpha!} (\Delta\bar{\beta}_{1i}^n)^{\alpha_1} (\Delta\bar{\beta}_{0i}^n)^{\alpha_2} (\Delta\bar{\eta}_i^n)^{\alpha_3},$$

for any $i \in [k]$, $\alpha_1 \in \mathbb{N}^d$, $\alpha_2 \in \mathbb{N}$ and $\alpha_3 \in \mathbb{N}^q$ such that $|\alpha_1| + \alpha_2 + |\alpha_3| = 1$.

**Step 2 - Non-vanishing coefficients:** In this step, we show that at least one among the ratios $S_{i,\alpha_1,\alpha_2,\alpha_3}^n/\mathcal{D}_{3n}$ does not go to zero as $n$ tends to infinity. Assume by contrary that all of them converge to zero, i.e. $S_{i,\alpha_1,\alpha_2,\alpha_3}^n/\mathcal{D}_{3n} \to 0$ as $n \to \infty$, for any $i \in [k]$, $\alpha_1 \in \mathbb{N}^d$, $\alpha_2 \in \mathbb{N}$ and $\alpha_3 \in \mathbb{N}^q$ such that $|\alpha_1| + \alpha_2 + |\alpha_3| = 1$. Then, it follows that

- $\frac{1}{\mathcal{D}_{3n}} \cdot \sum_{i=1}^{k} \|\Delta\bar{\beta}_{1i}^n\|_1 = \frac{1}{\mathcal{D}_{3n}} \cdot \sum_{i=1}^{k} \sum_{u=1}^{d} |S_{i,e_{d,u},0,\mathbf{0}_q}^n| \to 0$;

- $\frac{1}{\mathcal{D}_{3n}} \cdot \sum_{i=1}^{k} |\Delta\bar{\beta}_{0i}^n| = \frac{1}{\mathcal{D}_{3n}} \cdot \sum_{i=1}^{k} |S_{i,0_d,1,\mathbf{0}_q}^n| \to 0$;

- $\frac{1}{\mathcal{D}_{3n}} \cdot \sum_{i=1}^{k} \|\Delta\bar{\eta}_i^n\|_1 = \frac{1}{\mathcal{D}_{3n}} \cdot \sum_{i=1}^{k} \sum_{v=1}^{q} |S_{i,0_d,0,e_{q,v}}^n| \to 0$.

By taking the summation of the above three limits, we deduce that

$$\frac{1}{\mathcal{D}_{3n}} \cdot \sum_{i=1}^{k} \left[ |\Delta\bar{\beta}_{0i}^n| + \|\Delta\bar{\beta}_{1i}^n\|_1 + \|\Delta\bar{\eta}_i^n\|_1 \right] \to 0.$$

Due to the topological equivalence between the 1-norm and 2-norm, we achieve that

$$1 = \frac{1}{\mathcal{D}_{3n}} \cdot \sum_{i=1}^{k} \left[ |\Delta\bar{\beta}_{0i}^n| + \|\Delta\bar{\beta}_{1i}^n\| + \|\Delta\bar{\eta}_i^n\| \right] \to 0,$$

as $n \to \infty$, which is a contradiction. Thus, at least one among the ratios $S_{i,\alpha_1,\alpha_2,\alpha_3}^n/\mathcal{D}_{3n}$ must not approach zero as $n \to \infty$.

**Step 3 - Application of Fatou's lemma:** In this step, we use the Fatou's lemma to argue against the result in Step 2, and then, achieve the desired inequality in equation (35).

In particular, let us denote by $m_n$ the maximum of the absolute values of $S_{i,\alpha_1,\alpha_2,\alpha_3}^n/\mathcal{D}_{3n}$. Since at least one among those ratios must not approach zero as $n \to \infty$, we get that $1/m_n \not\to \infty$ as $n \to \infty$.

Recall from the hypothesis in equation (36) that $\|f_{G_n} - f_{\overline{G}}\|_{L^2(\mu)}/\mathcal{D}_{3n} \to 0$ as $n \to \infty$, which indicates that $\|f_{G_n} - f_{\overline{G}}\|_{L^1(\mu)}/\mathcal{D}_{3n} \to 0$ due to the equivalence between $L^1(\mu)$-norm and $L^2(\mu)$-norm. By means of the Fatou's lemma, we have

$$0 = \lim_{n\to\infty} \frac{\|f_{G_n} - f_{\overline{G}}\|_{L^1(\mu)}}{m_n \mathcal{D}_{3n}} \geq \int \liminf_{n\to\infty} \frac{|f_{G_n}(x) - f_{\overline{G}}(x)|}{m_n \mathcal{D}_{3n}} \mathrm{d}\mu(x) \geq 0.$$

This result implies that $[f_{G_n}(x) - f_{\overline{G}}(x)]/[m_n\mathcal{D}_{3n}] \to 0$ for almost every $x$.

Let us denote $S_{i,\alpha_1,\alpha_2,\alpha_3}^n/m_n\mathcal{D}_{3n} \to s_{i,\alpha_1,\alpha_2,\alpha_3}$ as $n \to \infty$ with a note that at least one among the limits $s_{i,\alpha_1,\alpha_2,\alpha_3}$ is non-zero. Then, it follows from the decomposition in equation (38) that

$$\sum_{i=1}^{k} \sum_{|\alpha|=1} s_{i,\alpha_1,\alpha_2,\alpha_3} \cdot \frac{\partial^{|\alpha_1|+\alpha_2}\sigma}{\partial\beta_1^{\alpha_1}\partial\beta_0^{\alpha_2}}(x, \bar{\beta}_{1i}, \bar{\beta}_{0i}) \frac{\partial^{|\alpha_3|}h}{\partial\eta^{\alpha_3}}(x, \bar{\eta}_i) = 0,$$

for almost every $x$. Note that the expert function $h(\cdot, \eta)$ is ..., then the above equation implies that $s_{i,\alpha_1,\alpha_2,\alpha_3} = 0$, for any $i \in [k]$, $\alpha_1 \in \mathbb{N}^d$, $\alpha_2 \in \mathbb{N}$ and $\alpha_3 \in \mathbb{N}^q$ such that $|\alpha_1| + \alpha_2 + |\alpha_3| = 1$. This contradicts the fact that at least one among the limits $s_{i,\alpha_1,\alpha_2,\alpha_3}$ is different from zero.

Hence, we obtain the local inequality in equation (35).

# B   Identifiability of the Sigmoid Gating Mixture of Experts

In this appendix, we explore the identifiability of the sigmoid gating MoE model in Proposition 1.

**Proposition 1.** *If the equation $f_G(x) = f_{G_*}(x)$ holds true for almost every $x$, then it follows that $G \equiv G'$.*

*Proof of Proposition 1.* From the assumption of this proposition, we have

$$\sum_{i=1}^{k} \frac{1}{1 + \exp(-\beta_{1i}^\top x - \beta_{0i})} \cdot h(x, \eta_i) = \sum_{i=1}^{k_*} \frac{1}{1 + \exp(-\beta_{1i}^{*\top} x - \beta_{0i}^*)} \cdot h(x, \eta_i^*). \tag{39}$$

Since the expert function $h(\cdot, \eta)$ is identifiable, the set $\{h(x, \eta_i') : i \in [k']\}$, where $\eta_1', \dots, \eta_{k'}'$ are distinct vectors for some $k' \in \mathbb{N}$, is linearly independent. If $k \neq k_*$, then there exists some $i \in [k]$ such that $\eta_i \neq \eta_j^*$ for any $j \in [k_*]$. This implies that $\frac{1}{1+\exp(-\beta_{1i}^\top x - \beta_{0i})} = 0$, which is a contradiction. Thus, we must have that $k = k_*$. As a result,

$$\left\{ \frac{1}{1 + \exp(-\beta_{1i}^\top x - \beta_{0i})} : i \in [k] \right\} = \left\{ \frac{1}{1 + \exp(-\beta_{1i}^{*\top} x - \beta_{0i}^*)} : i \in [k_*] \right\},$$

for almost every $x$. WLOG, we may assume that

$$\frac{1}{1 + \exp(-\beta_{1i}^\top x - \beta_{0i})} = \frac{1}{1 + \exp(-\beta_{1i}^{*\top} x - \beta_{0i}^*)}, \tag{40}$$

for almost every $x$ for any $i \in [k_*]$. It is worth noting that the sigmoid function is invariant to translations, then equation (40) indicates that $\beta_{1i} = \beta_{1i}^* + v_1$ and $\beta_{0i} = \beta_{0i}^* + v_0$ for some $v_1 \in \mathbb{R}^d$ and $v_0 \in \mathbb{R}$. Nevertheless, due to the assumptions $\beta_{1k} = \beta_{1k}^* = 0_d$ and $\beta_{0k} = \beta_{0k}^* = 0$, we have that $v_1 = 0_d$ and $v_0 = 0$. Consequently, it follows that $\beta_{1i} = \beta_{1i}^*$ and $\beta_{0i} = \beta_{0i}^*$ for any $i \in [k_*]$. Then, equation (39) can be represented as

$$\sum_{i=1}^{k_*} \frac{1}{1 + \exp(-\beta_{1i}^\top x - \beta_{0i})} \cdot h(x, \eta_i) = \sum_{i=1}^{k_*} \frac{1}{1 + \exp(-(\beta_{1i}^*)^\top x - \beta_{0i}^*)} \cdot h(x, \eta_i^*), \qquad (41)$$

for almost every $x$. Next, let us denote $J_1, J_2, \ldots, J_m$ as a partition of the index set $[k_*]$, where $m \leq k$, such that $(\beta_{0i}, \beta_{1i}) = (\beta_{0i'}^*, \beta_{1i'}^*)$ for any $i, i' \in J_j$ and $j \in [k_*]$. On the other hand, when $i$ and $i'$ do not belong to the same set $J_j$, we let $(\beta_{0i}, \beta_{1i}) \neq (\beta_{0i'}, \beta_{1i'})$. Thus, we can reformulate equation (41) as

$$\sum_{j=1}^{m} \sum_{i \in J_j} \frac{1}{1 + \exp(-\beta_{1i}^\top x - \beta_{0i})} \cdot h(x, \eta_i) = \sum_{j=1}^{m} \sum_{i \in J_j} \frac{1}{1 + \exp(-\beta_{1i}^{*\top} x - \beta_{0i}^*)} \cdot h(x, \eta_i^*),$$

for almost every $x$. Recall that $\beta_{1i} = \beta_{1i}^*$ and $\beta_{0i} = \beta_{0i}^*$ for any $i \in [k_*]$, then the above leads to

$$\{\eta_i : i \in J_j\} \equiv \{\eta_i^* : i \in J_j\},$$

for any $j \in [m]$. As a result, we achieve that

$$G = \sum_{j=1}^{m} \sum_{i \in J_j} \frac{1}{1 + \exp(-\beta_{0i})} \cdot \delta_{(\beta_{1i}, \eta_i)} = \sum_{j=1}^{m} \sum_{i \in J_j} \frac{1}{1 + \exp(-\beta_{0i}^*)} \cdot \delta_{(\beta_{1i}^*, \eta_i^*)} = G_*.$$

Hence, the proof is totally completed. □

## C   Additional Experiments

In this appendix, we conduct a simulation study to empirically validate our theoretical results on the convergence rates of the least squares estimators under both the Regime 1 and the Regime 2 of the sigmoid gating MoE. All the subsequent experiments are conducted on a MacBook Air equipped with an M1 chip CPU.

**Regime 1.** We begin with numerical experiments for the Regime 1 in which we sample synthetic data based on the model described in equation (1). In particular, we generate $\{(X_i, Y_i)\}_{i=1}^{n} \subset \mathbb{R}^d \times \mathbb{R}$ by first sampling $X_i \sim \mathrm{Uniform}([-1, 1]^d)$ for $i = 1, \ldots, n$. Then, we generate $Y_i$ according to the following model:

$$Y_i = f_{G_*}(X_i) + \epsilon_i, \quad i = 1, \ldots, n, \qquad (42)$$

where the regression function $f_{G_*}(\cdot)$ is defined as:

$$f_{G_*}(x) := \sum_{i=1}^{k_*} \frac{1}{1 + \exp(-(\beta_{1i}^*)^\top x - \beta_{0i}^*)} \cdot \varphi\left((a_i^*)^\top x + b_i^*\right), \qquad (43)$$

The input data dimension is set at $d = 32$. We employ $k_* = 8$ experts of the form $\varphi\left((a_i^*)^\top x + b_i^*\right)$, where the activation function is either the $\mathrm{ReLU}$ function or the identity function. The variance of Gaussian noise $\epsilon_i$ is specified as $\nu = 0.01$.

The ground-truth gating parameters $\beta_{0i}^* \in \mathbb{R}$ are drawn independently from an isotropic Gaussian distribution with zero mean and variance $\nu_g = 0.01/d$ for $1 \leq i \leq 6$, while we set $\beta_{11}^* = \ldots = \beta_{1k_*}^* = 0_d$. Similarly, the true expert parameters, $(a_i^*, b_i^*) \in \mathbb{R}^d \times \mathbb{R}$, are drawn independently of an isotropic Gaussian distribution with zero mean and variance $\nu_e = 1/d$ for all experts. These parameters remain unchanged for all the experiments (see also Table 3).

**Training procedure.** For each sample size $n$, spanning from $10^3$ to $10^5$, we perform 20 experiments. In every experiment, we employ $k = k_* + 1 = 9$ fitted experts, and the parameters initialization for the gating's and experts' parameters are adjusted to be near the true parameters, minimizing potential

Table 3: Ground-truth Parameters for Gating and Experts. The variance for the gating parameters is $\nu_g = 0.01/d$, and for the expert parameters is $\nu_e = 1/d$.

| Gating parameters | | Expert parameters | |
|---|---|---|---|
| $\beta_{1i}^* = 0_d$ | $\beta_{0i}^* \sim \mathcal{N}(0, \nu_g)$ | $a_i^* \sim \mathcal{N}(0_d, \nu_e I_d)$ | $b_i^* \sim \mathcal{N}(0, \nu_e)$ |

Table 4: True Parameters for Gating and Experts. The variance for the gating parameters is $\nu_g = 0.01/d$ and for the expert parameters is $\nu_e = 1/d$.

| Gating parameters | | Expert parameters | | |
|---|---|---|---|---|
| $\begin{cases} \beta_{1i}^* \sim \mathcal{N}(0_d, \nu_g I_d) & 1 \le i \le 7 \\ \beta_{1i}^* = 0_d & i = 8 \end{cases}$ | $\beta_{0i}^* \sim \mathcal{N}(0, \nu_g)$ | | $a_i^* \sim \mathcal{N}(0_d, \nu_e I_d)$ | $b_i^* \sim \mathcal{N}(0, \nu_e)$ |

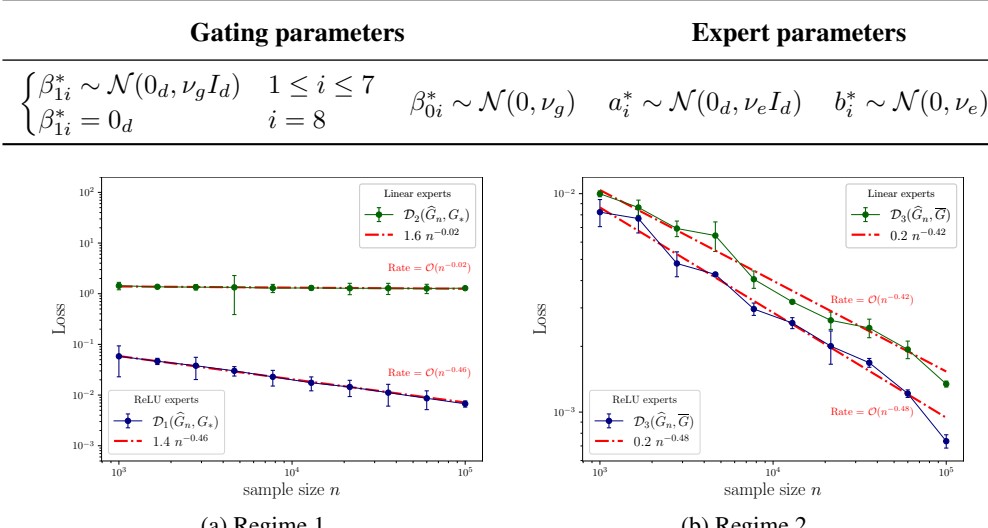

(a) Regime 1  (b) Regime 2

Figure 2: Logarithmic plots of empirical convergence rates. Figures 2a and 2b illustrate the empirical averages of the corresponding Voronoi losses under the Regime 1 and the Regime 2, respectively. The blue lines depict the Voronoi loss associated with the ReLU experts, while the green lines correspond to that of the linear experts. The red dash-dotted lines are used to illustrate the fitted lines for determining the empirical convergence rates. See Appendix C for the experimental details.

instabilities from the optimization process. Subsequently, we execute the stochastic gradient descent algorithm across 10 epochs, employing a learning rate of $\eta = 0.1$ to fit a model to the synthetic data.

**Results.** For each experiment, we calculate the Voronoi losses for every model and report the mean values for each sample size in Figure 2a. Error bars representing two standard deviations are also shown. In Figure 2a, both the empirical convergence rates corresponding to the ReLU experts and the linear experts are analyzed under the over-specified setting. It can be seen that the use of ReLU experts induces a rapid convergence rate of order $\mathcal{O}(n^{-0.46})$, while the linear experts lead to a considerably slower rate of order $\mathcal{O}(n^{-0.02})$. Those empirical rates match our theoretical results in Theorem 2 and Theorem 4, respectively.

**Regime 2.** In the experiments for the Regime 2, we apply the same experimental setup as in those for the Regime 1 except for the generation of true parameters $\beta_{1i}^*$. More specifically, $\beta_{1i}^*$ are drawn independently from an isotropic Gaussian distribution $\mathcal{N}(0_d, \nu_g I_d)$ for $1 \le i \le 8$, where $\nu_g = 0.01/d$, while we set $\beta_{1i}^* = 0_d$ for $i = 8$ (see also Table 4).

**Results.** It can be observed from Figure 2b that the uses of ReLU experts and linear experts both lead to fast empirical convergence rates of orders $\mathcal{O}(n^{-0.48})$ and $\mathcal{O}(n^{-0.42})$, respectively. Those empirical rates align with our theoretical findings in Theorem 5.

