# OpenReview forum: "Sigmoid Gating is More Sample Efficient than Softmax Gating in Mixture of Experts"
_NeurIPS.cc/2024/Conference — NeurIPS 2024 poster_

### Official Review · Reviewer_4kwq · 2024-07-11

**Soundness:** 2
**Presentation:** 2
**Contribution:** 2
**Rating:** 6
**Confidence:** 3

**Summary:**

This paper provides a theoretical analysis of the convergence rate of the least squares estimator for learning MoE with sigmoid gating. Based on the results, the authors conclude that sigmoid gating enjoys a faster convergence rate and requires a smaller sample size to achieve the same error compared to softmax gating.

**Strengths:**

1. The topic is very interesting. It is intriguing to see how the selection of gating functions can affect sample efficiency in MoE.
2. This problem is particularly relevant to recent important topics regarding language models, given that MoE has been applied in popular models like Mixtral.

**Weaknesses:**

1. The biggest concern to me is that the results from this paper do not fully support the main claim that “sigmoid gating is more sample efficient than softmax gating”. If I understand correctly (correct me if I'm wrong), the comparison between sigmoid and softmax is not under the same scenario. The convergence rate for softmax in Table 1 is from the analysis in [21], where the ground truth is an MoE model with softmax gating, while this paper considers the setting where the ground truth is an MoE model with sigmoid gating. These are two different setups and do not necessarily imply that, given the same ground truth function, sigmoid is necessarily more sample efficient than softmax.

2. The only existing empirical work discussing the potential superiority of sigmoid gating that the authors have mentioned is [3], where the provided intuition is that softmax introduces competition between experts, while sigmoid does not have this issue. This is very straightforward, although not yet formally and theoretically justified. On the other hand, this paper explains the advantage from the perspective of sample efficiency. I wonder how these two perspectives can be connected. Is there anything regarding the competition view that can be reflected in this paper’s analysis? I believe this is important for understanding the effect of sigmoid gating.

3. Following point 2, it would be helpful to provide empirical results to further support the main conclusion that sigmoid gating is more sample efficient. Even experiments on toy data can help. Otherwise, the statement regarding sample efficiency lacks empirical evidence. Additionally, even "sigmoid gating is better than softmax gating" alone does not seem to be well-supported by empirical observations from the literature.

**Questions:**

See my questions raised in Weaknesses.

**Limitations:**

The authors have discussed certain limitations in Section 4.

---

> ### Author Rebuttal · Authors · 2024-08-05
>
> ### **Q1: The results from this paper do not fully support the main claim that “sigmoid gating is more sample efficient than softmax gating” because the comparison between sigmoid and softmax is not under the same setup.**
>
> Thanks for your comment. Let us explain why it is reasonable to compare the expert estimation rates under the MoE model when using the sigmoid gating versus when using the softmax gating.
>
> Our paper (resp. Nguyen et. al. [21]) consider a well-specified setting where the data are generated from a regression model where the regression function is a sigmoid gating (resp. softmax gating) MoE in order to lay the foundation for a more realistic yet challenging misspecified setting where the data are not necessarily generated from those models.
>
> Under that misspecified setting, the regression function is an arbitrary function $g(x)$ which is not necessarily a mixture of experts. Then, the least square estimator $\widehat{G}_n$ defined in equation (3) converges to the mixing measure $\widetilde{G}\in\mathcal{G}_k$ that minimizes the distance
>
> $||f_{G}-g||_2$,
>
> where $f_{G}$ is the regression function taking the form of the sigmoid gating (resp. softmax gating) MoE.
>
> The insights from our theories and from Nguyen et. al. [21] under the well-specified setting indicate that the Voronoi loss functions can be used to obtain the estimation rates of individual parameters of the least square estimator $\widehat{G}_n$ to those of $\widetilde{G}$, and therefore, the expert estimation rates. Let us recall from Table 1 that under the Regime 1 (all the over-specified gating parameters are zero), using the sigmoid gating leads to the same expert estimation rates as when using the softmax gating. However, under the Regime 2 (not all the over-specified gating parameters equal zero), which is more likely to occur, the sigmoid gating totally outperforms the softmax gating in terms of the rates for estimating feed-forward expert networks and polynomial experts ($n^{-1/2}$ compared to either $n^{-1/4}$ or $1/\log(n)$).
>
> Thus, under the misspecified setting, the expert estimation rates achieved when we use the sigmoid gating should be faster than those obtained when we use the softmax gating. This accounts for the validity of the expert estimation rate comparison between these two gating functions.
>
> ### **Q2: Is there anything regarding the non-competition of the sigmoid gating that can be reflected in the analysis?**
>
> Thanks for your question. Let us show the connection between the non-competition of the sigmoid gating and our analysis. In particular, when using the sigmoid gating, the experts do not share the gating parameters, that is, the mixture weights are independent of each other. Thus, the interaction between the gating and expert parameters which induces slow estimation rates when using the softmax gating (see Eq. (4) in [21]) does not hold if we use the sigmoid gating. As a result, the expert estimation rates when using the sigmoid gating are either comparable (under the Regime 1) or faster (under the Regime 2) than those when using the softmax gating.
>
> ### **Q3: It would be helpful to provide empirical results to further support the main conclusion that sigmoid gating is more sample efficient. Even experiments on toy data can help.**
>
> Thanks for your suggestion. Please refer to our response to the Common Question 1 in the General Section for further details.
>
> ### **Q4: Even "sigmoid gating is better than softmax gating" alone does not seem to be well-supported by empirical observations from the literature.**
>
> Thanks for your comment. However, we respectfully disagree that the claim "sigmoid gating is better than softmax gating" is not well-supported by empirical observations from the literature. In particular, there are two recent works [1, 3] on the applications of MoE in language modeling showing that the performances when using the sigmoid gating are comparable or even better than those when using the softmax gating.
>
> Finally, we would like to emphasize that our main goal is to demonstrate that the sigmoid gating is more sample efficient than the softmax gating from the perspective of the expert estimation problem. We have not attempted to show that the sigmoid gating is better than the softmax gating in general.
>
> **References**
>
> [1] Z. Chi. On the representation collapse of sparse mixture of experts. Advances in NeurIPS, 2022.
>
> [3] R. Csordás. Approximating two-layer feedforward networks for efficient transformers. Findings of the EMNLP 2023.
>
> [21] H. Nguyen. On least squares estimation in softmax gating mixture of experts. In ICML, 2024.

---

> > ### Comment · Reviewer_4kwq · 2024-08-12
> >
> > Thank you to the authors for their responses. However, I am not entirely sure how the authors' response addresses my first question. My question was: this paper and Nguyen et al. [21] consider different settings. This paper considers the case where the data are generated with sigmoid gating, and a model with sigmoid gating is trained to fit the data; Nguyen et al. [21] consider the case where the data are generated with softmax gating, and a model with softmax gating is trained to fit the data. Since the data in these two cases are not generated in the same way, we cannot claim that sigmoid gating is more sample efficient than softmax if we use them to fit the same data. In other words, if the paper had shown that, given the same data generated in a certain way, using sigmoid to fit the data is more sample efficient than using softmax, then I would find the conclusion well justified.
> >
> > The new empirical results seem to be done in the desired setting, where the data are always generated with softmax gating, whether using the sigmoid gating model or the softmax gating model to fit the data.

---

> ### Author Response · Authors · 2024-08-13
> **Response to Reviewer 4kwq**
>
> Dear Reviewer 4kwq,
>
> We would like to thank you for raising your concerns. **We hope that our following response will address those concerns, and eventually convince you to increase your rating**.
>
> Per your suggestion, we conduct both theoretical and empirical sample efficiency comparisons between the sigmoid gating and the softmax gating under the setting where the data are generated from the same source. Below are the expert estimation rates when using the softmax gating MoE and the sigmoid gating MoE to fit the data, respectively.
>
> **Same data generation setting:** The data $(X_1,Y_1),\ldots,(X_n,Y_n)$ are generated from a regression framework
> $$Y_i=g(X_i)+\epsilon_i,$$
> where the features $X_1,\ldots,X_n$ are sampled from a probability distribution $\mu$, and $\epsilon_1,\ldots,\epsilon_n$ are independent Gaussian noise variables such that $\mathbb{E}[\epsilon_i|X_i]=0$ and $Var(\epsilon_i|X_i)=\nu$ for all $1\leq i\leq n$. The unknown regression function $g(x)$ is not necessarily a mixture of experts (MoE). Then, we use either the sigmoid gating MoE or the softmax gating MoE to fit the data.
>
> In particular, the least square estimator $\widehat{G}_n$ in Eq.(3) now converges to a mixing measure $\widetilde{G}\in\mathcal{G}_k(\Theta)$ where
>
> $$\widetilde{G}\in\arg\min_{G}||f_{G}-g||_2,$$
>
> in which $f_{G}$ is the regression taking the form of the MoE associated with either the sigmoid gating or the softmax gating. Below are the expert estimation rates resulted from our convergence analysis:
>
> **When $f_{G}$ is the softmax gating MoE:** the rates for estimating feed-forward expert networks with ReLU activation are of order $n^{-1/4}$, while those for polynomial experts are slower than any polynomial rates, and could be as slow as $1/\log(n)$;
>
> **When $f_{G}$ is the sigmoid gating MoE:** the estimation rates for feed-forward expert networks with ReLU activation and polynomial experts share the same order of $n^{-1/2}$, which are significantly faster than those when using the softmax gating MoE.
>
> **Empirical validation:** To justify the above theoretical results, we conduct a simulation study in the General Response section where we take polynomial experts into account. The empirical result totally matches its theoretical rates.
>
> As a consequence, **we can conclude that the sigmoid gating is more sample efficient than the softmax gating even when fitting the same data**.
>
> Please feel free to let us know if you have any further concerns regarding the paper. We are more than happy to address all of them.
>
> Best,
>
> The Authors

---

> > ### Comment · Reviewer_4kwq · 2024-08-13
> >
> > Thank the authors for their response. I’m curious about how the new theoretical results presented relate to those in the original paper. Are they completely independent, an extension of the original findings, or already implied by them? If they are a more generalized version of the original results, do we need to introduce any new assumptions here, particularly any assumptions on $g(x)$?

---

> ### Author Response · Authors · 2024-08-13
>
> Dear Reviewer 4kwq,
>
> Thanks for your question. We would like to confirm that the analysis of the sigmoid gating (resp. softmax gating) MoE under the same data generation setting is implied by the analysis in our paper (resp. in [a]).
>
> **Assumption on the regression function $g$.** Firstly, let us recall a result on the universal approximation of the sigmoid function in [b]. In particular, let $g:\mathcal{X}\to\mathbb{R}$ be a function such that there is a Fourier representation of the form
>
> $$g(x)=\int_{\mathcal{X}}e^{i\omega\cdot x}\tilde{g}(\omega)d\omega,$$
>
> for some complex-valued function $\tilde{g}(x)$ for which $\omega\tilde{g}(\omega)$ is integrable and the term
>
> $$C_g=\int_{\mathcal{X}}||\omega||_2|\tilde{g}(\omega)|d\omega$$
>
> is finite. Then, there exists a linear combination of $\tilde{k}$ sigmoidal functions $f_{\widetilde{G}}(x)$ such that
>
> $$\int_{\mathcal{X}}[f_{\widetilde{G}}(x)-g(x)]^2d\mu(x)\leq\frac{(2C_g)^2}{\tilde{k}}.$$
>
> **Analysis for the sigmoid gating MoE.** Subsequently, we combine the above result with our current analysis in the paper. More specifically, by treating the mixing measure $\widetilde{G}$ as the mixing measure $G_*$ in the paper, we are able to design a Voronoi loss
>
> $$\mathcal{D}(G,\widetilde{G})=\sum_{j=1}^{k}\sum_{i\in\mathcal{A}_j}$$
>
> $$\Big[|\beta_{0i}-\tilde{\beta}_{0j}|$$
>
> $$+||\beta_{1i}-\tilde{\beta}_{1j}||$$
>
> $$+||\eta_{i}-\tilde{\eta}_{j}||\Big],$$
>
> and show that
>
> $$\mathcal{D}(\widehat{G}_n,\widetilde{G})=O_P(\sqrt{\log(n)/n}).$$
>
> From this bound, we deduce that the expert estimation rates are of order $O_P(n^{-1/2})$.
>
> **Analysis for the softmax gating MoE.** Similarly, we also leverage the universal approximation of the softmax function in [c], and the analysis in [a] to derive the expert estimation rates under the same data generation setting. In particular, we find out that the rates for estimating feed-forward expert networks with ReLU activation are of order $n^{-1/4}$, while those for polynomial experts are slower than any polynomial rates, and could be as slow as $1/\log(n)$.
>
> **As the discussion period deadline is approaching, please let us know if you have any further concerns. We are more than happy to answer your questions. Additionally, if you find that our response sufficiently addresses your concerns, we hope that you will re-evaluate the paper and increase the rating. Thank you again!**
>
> **References**
>
> [a] H. Nguyen. On least squares estimation in softmax gating mixture of experts. In ICML, 2024.
>
> [b] Andrew R. Barron. Universal Approximation Bounds for Superpositions of a Sigmoidal Function. IEEE Transactions on Information Theory, 1993.
>
> [c] Assaf J. Zeevi. Error Bounds for Functional Approximation and Estimation Using Mixtures of Experts. IEEE Transactions on Information Theory, 1998.

---

> > ### Comment · Reviewer_4kwq · 2024-08-13
> >
> > Thank the authors for addressing my concerns. I believe the step of connecting the two data generation processes with the universal approximation properties of the sigmoid and softmax functions is essential in clarifying and legitimizing the comparison. It would be beneficial to include this in the revised version.
> >
> > I have adjusted my score accordingly.

---

> > > ### Author Response · Authors · 2024-08-13
> > > **Thank You!**
> > >
> > > Dear Reviewer 4kwq,
> > >
> > > We would like to thank you for increasing your rating to 6, we really appreciate that. We will include the connection of the universal approximation properties of the sigmoid and softmax functions and the derivation of expert estimation rates under the same data generation setting in the revision of our paper.
> > >
> > > Best,
> > >
> > > The Authors

---

### Official Review · Reviewer_tkzC · 2024-07-16

**Soundness:** 3
**Presentation:** 2
**Contribution:** 2
**Rating:** 6
**Confidence:** 3

**Summary:**

The paper argues that the sigmoid gating function is more sample efficient than the softmax gating function in mixture of experts (MoE) modeling. It removes competition and estimates the contribution each expert independently.

Empirical studies show that sigmoid gating achieves superior performance, but the paper aims to provide theoretical backing for this claim.

They consider a regression framework and analyze the rates of convergence of the least squares estimator in over-specified cases.

The convergence rates for expert estimation are derived under two regimes: Regime 1 (all over-specified parameters are 0) and Regime 2 (at least one over-specified parameter is not 0).

**Strengths:**

The paper demonstrates that sigmoid gating is more sample efficient than softmax gating, requiring fewer samples to achieve the same level of accuracy in expert estimation.

experts with feed-forward networks and commonly used activations (ReLU, GELU) have faster convergence rates under sigmoid gating than softmax gating.

The sigmoid gating mechanism is compatible with a broader class of expert functions compared to softmax gating. (ReLU and GELU, as well as polynomial activations)

**Weaknesses:**

The results are heavily dependent on specific assumptions, such as the distinctness of expert parameters and the boundedness of the input space. If these assumptions are violated in practical scenarios, the theoretical guarantees may not hold

While the derivations are very appropriate and i appreciate the citations, it would be interesting to see even toy experiments byt the authorsthat show similar conversions to the theoretical ones.

**Questions:**

the paper lacks justification for the choice of the regimes. We also need to deep dig to understand the reasons for strong identifiability conditions (for experts to be distinct enough, if i understand correctly), bracketing entropy (model is not overly complex) and concentration inequalities. it is quite difficult to read, so some more intuition and explanation would be helpful.

Minor:
124 in can seen

**Limitations:**

As mentioned by the authors,

The paper assumes that the ground-truth parameters are independent of the sample size, leading to point-wise rather than uniform estimation rates.

The assumption that the true regression function belongs to the parametric class of MoE models under sigmoid gating is restrictive. This assumption is likely to be violated in real-world settings.

I still find these assumptions reasonable.

---

> ### Author Rebuttal · Authors · 2024-08-05
>
> ### **Q1: The results are heavily dependent on specific assumptions, such as the distinctness of expert parameters and the boundedness of the input space. If these assumptions are violated in practical scenarios, the theoretical guarantees may not hold.**
>
> Thanks for your comments. We will explain why the assumptions mentioned above are reasonable.
>
> Firstly, let us begin with the assumption of distinct expert parameters. This assumption is to ensure that all the experts are different from each other. In practice, it is memory-inefficient to have two identical experts as they have the same expertise but we have to store different weight parameters for them. If there are two identical experts, we can merge them by taking the summation of their weights.
>
> Secondly, regarding the assumption of the bounded input space, we would like to emphasize that this is a standard assumption in the literature of expert estimation in the MoE (see [12, 21]). Moreover, we can address the issue that the magnitude of the input goes to infinity by normalizing the input value, which has been recently applied in practice (see [1, 36]) and should not affect the current theory.
>
> ### **Q2: It would be interesting to see even toy experiments by the authors that show similar conversions to the theoretical ones.**
>
> Thanks for your suggestion. Please refer to our response to the Common Question 1 in the General Section for further details.
>
> ### **Q3: The paper lacks justification for the choice of the regimes.**
>
> Thanks for your comment. Actually, we have already included the justification for dividing the convergence analysis into two regimes in the "Technical challenges" paragraph (see lines 83-90). Let us summarize it here.
>
> Firstly, we would like to emphasize that the regimes are determined based on the gating convergence. Recall that the true number of experts $k_*$ is unknown, and we over-specify the true model by a mixture of $k$ experts where $k>k_*$. Thus, there must be at least one atom of $G_*$ fitted by two atoms of $\widehat{G}_n$.
>
> WLOG, assume that $(\hat{\beta}^n_{1i},\hat{\eta}^n_i)\to(\beta^*_1,\eta^*_1)$ for all $i\in\{1,2\}$.
>
> Then, we have $h(x,\hat{\eta}^n_i)\to h(x,\eta^*_1)$ for all $i\in\{1,2\}$.
>
> Therefore, the regression function $f_{\widehat{G}_n}$ converges to
>
> $f_{G_*}$ only if
> $$\sum_{i=1}^{2}\sigma((\hat{\beta}^n_{1i})^{\top}x+\hat{\beta}^n_{0i})\to\sigma((\beta^*_{11})^{\top}x+\beta^*_{01}),$$
> for almost every $x$, where $\sigma$ denotes the sigmoid function. This above limit occurs iff $\beta^*_{11}=0_d$. As a consequence, we propose conducting the analysis under two following complement regimes:
>
> Regime 1: All the overspecified parameters $\beta^*_{1i}$ are equal to zero;
>
> Regime 2. At least one among the over-specified parameters $\beta^*_{1i}$ is different from zero.
>
> ### **Q4: What are the explanations for the strong identifiability condition?**
>
> Thanks for your question. We will explain the strong identifiability condition both intuitively and technically as follows:
>
> **Intuitively**, the strong identifiability condition helps eliminate potential interactions among parameters expressed in the language of PDE (see Eq. (8) and Eq. (11) where gating parameters $\beta_1$ interact with expert parameters $a$). Such interactions are demonstrated to result in significantly slow expert estimation rates (see Theorem 3 and Theorem 4).
>
> **Technically**, a key step in our proof techniques rely on the decomposition of the discrepancy between $f_{\widehat{G}_n}(x)$ and
>
> $f_{G_*}(x)$ into a combination of linearly independent terms. This can be done by applying Taylor expansions to the function $F(x,\beta_1,\beta_0,\eta):=\sigma(\beta_1^{\top}x+\beta_0)h(x,\eta)$ defined as the product of the sigmoid gating and the expert function $h$. Thus, the condition is to ensure that terms in the decomposition are linearly independent.
>
> ### **Q5: Typo issue.**
>
> Thanks for pointing out. We will correct them in the revision of our paper.
>
> **References**
>
> [1] Z. Chi. On the representation collapse of sparse mixture of experts. Advances in NeurIPS, 2022.
>
> [12] N. Ho. Convergence rates for Gaussian mixtures of experts. In JMLR, 2022.
>
> [21] H. Nguyen. On least squares estimation in softmax gating mixture of experts. In ICML, 2024.
>
> [36] B. Li. Sparse mixture-of-experts are domain generalizable learners. In ICLR, 2023.

---

> > ### Comment · Reviewer_tkzC · 2024-08-11
> >
> > Thank you for addressing the questions. As I am leaning towards acceptance, I raised my score to 6.

---

> > > ### Author Response · Authors · 2024-08-12
> > > **Thank You**
> > >
> > > Dear Reviewer tkzC,
> > >
> > > We want to thank you for increasing your score to 6.
> > >
> > > Please let us know if you still have any concerns about the paper.
> > >
> > > Best,
> > >
> > > The Authors

---

### Official Review · Reviewer_PmAq · 2024-07-18

**Soundness:** 3
**Presentation:** 2
**Contribution:** 2
**Rating:** 5
**Confidence:** 3

**Summary:**

This paper presents a theoretical analysis of expert estimation in MoE models using sigmoid gating, in contrast to the more widely used softmax gating. The authors show that sigmoid gating leads to better sample efficiency compared to softmax gating for estimating expert parameters. In particular, the paper analyzes convergence rates for least squares estimation under two regimes:
* Regime 1) when all over-specified gating parameters are zero
* Regime 2) when at least one of the over-specified gating parameters is non-zero

One of the key findings is that experts formulated as NNs with common activations like ReLU and GELU obtain faster convergence rates under sigmoid gating compared to softmax gating. This work provides theoretical justification for the potential empirical benefits of sigmoid gating, showing it requires smaller sample sizes to achieve the same estimation error as softmax gating.

**Strengths:**

- The paper provides rigorous proofs and establishes convergence rates under different conditions.
- Sigmoid gating function has been less explored in the MoE field and analyzing its behaviour for expert estimation problem is interesting.

**Weaknesses:**

1) The structure of the paper is very confusing and could be much improved. There is way too much repetition in the paper. For example, the section Technical challenges in the introduction which defines Regime 1 and 2 (Lines 83-92) is largely copy pasted to lines 155-165 and is redundant. The various sections in the introductions particularly the contributions should be kept at a more high level and not just copy paste the method section. The authors should provide a more concise overview of the problem and main contributions. Remove technical details that are repeated in later sections.

2) The recent insights into training Sparse MoEs demonstrate that "the common practice of setting the size of experts in MoE to mirror the feed-forward layer (of a base dense network) is not optimal at almost any computational budget" (Jakub Krajewski et al. 2024). If in practice, top-k combination of small fine-grained experts is proven to be more effective than a single large expert, are we dealing in a regime where all true experts are *over-specified* in practice? If this is the case what are the implications for Regime 1? Does it mean that none of the experts are input-dependent as all the over-specified parameters $β^*_{1i}$ are equal to $0_d$? Does Regime 1 hold in practice?

3) It would be best if the authors could find some practical implications of their findings. The theoretical work presented in the paper does not establish any relationship to any of the SoTA MoE models in the literature. It is unclear to me how impactful the convergence analysis with the considered regimes is in practical MoE research works.

4) At present, the content and organization of the paper is very close to the paper "On Least Square Estimation in Softmax Gating Mixture of Experts" (Nguyen et al. 2024). The sections are so similar that may cause copyright issues.

References:
1. Krajewski, Jakub, et al. "Scaling laws for fine-grained mixture of experts." arXiv preprint arXiv:2402.07871 (2024).

2. Nguyen, Huy, Nhat Ho, and Alessandro Rinaldo. "On least squares estimation in softmax gating mixture of experts." ICML (2024).

**Questions:**

Minor point:
- misspecified is misspelled in line 175.dq

**Limitations:**

Two limitations concerning the assumptions used in the analyses are discussed in the limitations section.

---

> ### Author Rebuttal · Authors · 2024-08-05
>
> ### **Q1: The authors should provide a more concise overview of the main contributions, and remove technical details that are repeated in later sections.**
>
> Thanks for your suggestions. We will modify the contribution paragraph as below, and consider removing repeated details in Section 2 in the revision.
>
> **Contributions.** In this paper, we carry out a convergence analysis of the sigmoid gating MoE under two regimes of the gating parameters. The main objective is to compare the sample efficiency between the sigmoid gating and the softmax gating. Our contributions can be summarized as follows:
>
> **(C.1) Convergence rate for the regression function.** We demonstrate in Theorem 1 that the regression estimation $f_{\widehat{G}_n}$
>
> converges to its true counterpart $f_{G_*}$ at the rate of order $\mathcal{O}_P(n^{-1/2})$, which is parametric on the sample size $n$. This regression estimation rate is then utilized for determining the expert estimation rates.
>
> **(C.2) Expert estimation rates under the Regime 1.** Under the first regime, we first establish a condition called *strong identifiability* to characterize which types of experts would yield polynomial estimation rates. In particular, we find out that the rates for estimating experts formulated as feed-forward networks (FFN) with popular activation functions such as ReLU and GELU are of polynomial orders. By contrast, those for polynomial experts and input-indepedent experts could be of order $\mathcal{O}_P(1/\log(n)$. Such expert convergence behavior is similar to that when using the softmax gating.
>
> **(C.3) Expert estimation rates under the Regime 2.** Under the second regime, the regression estimation $f_{\widehat{G}_n}$
>
> converge to a function taking the form of a sigmoid gating MoE which is different from $f_{G_*}$. From our derived weak identifiability condition, it follows that estimation rates for FFN experts with ReLU or GELU activation and polynomial experts are of orders $O_P(n^{-1/2})$, which are substantially faster than those when using the softmax gating (see Table 1). Therefore, the sigmoid gating is more sample efficient than the softmax gating.
>
> ### **Q2: If in practice, when using top-k combination of small fine-grained experts, are we dealing in a regime where all true experts are over-specified in practice? If this is the case what are the implications for Regime 1? Does Regime 1 hold in practice?**
>
> Thanks for your questions.
>
> Firstly, when using the regression estimation as a mixture of small fine-grained experts [34] (which could have up to millions of experts [35]), it is highly likely that we are dealing with the scenario when all the true experts are over-specified.
>
> Secondly, assume that we use a mixture of small fine-grained experts, and all the true experts are over-specified. Then, under the Regime 1, all the gating parameters $\beta^*_{1j}$ are equal to zero, which means that the true mixture weights (gating values) are independent of the input $x$. Moreover, the fitted parameters $\widehat{\beta}^n_{1i}$ must also converge to zero. Consequently, all the fitted mixture weights also become independent of the input $x$. However, according to the definitions of two regimes, the Regime 1 is much less likely to occur than the Regime 2 where at least one among the over-specified parameters $\beta^*_{1j}$ is different from zero.
>
> ### **Q3: What are the practical implications of the convergence analysis?**
>
> Thanks for your question. There are two important practical implications from our analysis:
>
> **(i) Expert specialization:** The convergence behavior of expert estimation allows us to capture how fast an expert learns a specific task, which is one of the most important problems in the MoE literature known as expert specialization (see [4]). As the sigmoid gating is more sample efficient than the softmax gating, our theories suggest that it would be better to use the sigmoid gating in this field.
>
> **(ii) Expert compatibility:** Compared to the softmax gating, the estimation rates for feed-forward expert networks with ReLU activation and polynomial experts when using the sigmoid gating are much faster. Thus, our theories indicate that the sigmoid gating is compatible with a broader class of experts than the softmax gating. This implication is particularly useful when people employ a mixture of fine-grained (shallow) expert networks [35].
>
> ### **Q4: The content and organization of the paper are very close to the paper [21].**
>
> Thanks for your comment. However, we respectfully disagree that the content and organization of our paper are close to those of [21] for the following reasons:
>
> **1. Content:**
>
> *(1.1) Different objectives:* the objective of [21] is to figure out what types of experts are compatible with the softmax gating in terms of estimating experts. Meanwhile, our paper focuses on the sample efficiency comparison between the sigmoid gating and the softmax gating
>
> *(1.2) Analysis of similar expert types for comparison*: since our main goal is to demonstrate that the sigmoid gating is more sample efficient than the softmax gating, it is necessary to analyze experts considered in [21] for the sake of comparison. This probably makes our content look similar to that in [21]. However, the derived expert estimation rates are different, particularly under the Regime 2 where the sigmoid gating outperforms its softmax counterpart (see Table 1).
>
> **2. Organization:** In [21], sections for main results are divided based on the types of experts, namely Section 3 for strongly identifiable experts, Section 4.1 for ridge experts with strongly independent activation function, and Section 4.2 for polynomial experts. On the other hand, in our paper, sections for main results are organized based on two regimes of gating parameters. More specifically, Section 3.1 is for Regime 1, while Section 3.2 is for Regime 2.
>
> **References**
>
> *Due to the space limit, we leave the references to the General Response section.*

---

> > ### Comment · Reviewer_PmAq · 2024-08-12
> >
> > Thanks for your detailed response. Most of my concerns are addressed. I would encourage the authors to improve the conciseness of the presentation in the revised version as discussed in the comments. I raise my rating to 5

---

> > > ### Author Response · Authors · 2024-08-12
> > > **Thank you!**
> > >
> > > Dear Reviewer PmAq,
> > >
> > > We would like to thank you for increasing the rating to 5. We will definitely incorporate the modifications into the revision of our paper as discussed. Please feel free to let us know if you have any further concerns.
> > >
> > > Best,
> > >
> > > The Authors

---

### Author Rebuttal · Authors · 2024-08-05

# **General Response**

Dear AC and reviewers,

We would like to thank you for your value feedback and constructive comments, which have helped us improve the paper substantially. We are encouraged by the endorsement that:

- Sigmoid gating function has been **less explored** in the MoE field, and the topic is **very interesting** (Reviewer PmAq and Reviewer 4kwq).

- This problem is particularly **relevant to recent important topics** regarding language models (Reviewer 4kwq).

- The paper provides **rigorous proofs** and establishes convergence rates under different conditions (Reviewer PmAq).

There is one common concerns from the reviewers regarding the synthetic experiments, which will be addressed in the sequel.

### **CQ1: Synthetic experiments for empirically justifying the theoretical results.**

Thanks for your suggestion. Actually, we already conducted a simulation study to empirically validate our theoretical results on the convergence rates of the least squares estimators under both the Regime 1 and the Regime 2 of the sigmoid gating MoE in Appendix C.

Moreover, we just also carry out the following numerical experiments which empirically demonstrate that the sigmoid gating is more sample efficient than softmax gating in MoE.

**Experimental setup.** From Table 1 in our paper, it can be seen that the sigmoid gating shares the same expert estimation rates as the softmax gating under the Regime 1. However, the former gating outperforms the latter in terms of expert estimation rates under the Regime 2, particularly for polynomial experts. Therefore, we will consider linear experts and the Regime 2 in our subsequent experiments. Due to the time limit, we will include additional experiments for other setups in the revision later.

In particular, we generate the data by first sampling $X_i \sim \mathrm{Uniform}([-1, 1]^d)$ for $i = 1, \ldots, n$. Then, we generate $Y_i$ according to the following model:

$$Y_{i} = g_{G_{*}}(X_{i}) + \epsilon_{i},$$

where the regression function $g_{G_{*}}(\cdot)$ take the form of a softmax gating MoE:

$$\sum_{i=1}^{k_*} softmax((\beta^*_{1i})^{\top}x+\beta^*_{0i})\cdot \left((a_i^*)^\top x + b_{i}^{*}\right).$$

The input data dimension is $d = 32$. We employ $k_* = 8$ experts of the form $a^{\top}x+b$. The variance of Gaussian noise $\epsilon_i$ is $\nu = 0.01$.

The ground-truth gating parameters $\beta^*_{0i}$ are drawn independently from an isotropic Gaussian distribution with zero mean and variance $\nu_g = 0.01/d$ for $1 \le i \le 6$, while $\beta^*_{1i}$ are drawn independently from an isotropic Gaussian distribution $\mathcal{N}(0_d, \nu_g I_{d})$ for $1\leq i\leq 7$, where $\nu_g=0.01/d$, and we set $\beta^*_{1i}=0_d$ for $i=8$. Similarly, the true expert parameters, $(a_i^*, b_i^*)$, are drawn independently of an isotropic Gaussian distribution with zero mean and variance $\nu_e = 1/d$ for all experts.

**Training procedure.** For each sample size $n$, spanning from $10^3$ to $10^5$, we perform 20 experiments. In every experiment, we employ $k=k_*+1=9$ fitted experts, and the parameters initialization for the gating's and experts' parameters are adjusted to be near the true parameters, minimizing potential instabilities from the optimization process. Subsequently, we execute the stochastic gradient descent algorithm across $10$ epochs, employing a learning rate of $\eta = 0.1$ to fit a model to the synthetic data.

**Results.** For each experiment, we calculate the Voronoi losses for every model and report the mean values for each sample size in Figure 1 in the attached PDF file. Error bars representing two standard deviations are also shown.
In Figure 1, the Voronoi losses associated with the sigmoid gating vanish at the rate of $O(n^{-0.4})$, which nearly matches our theoretical results in Theorem 5 in our paper. Meanwhile, those associated with the softmax gating converge to zero at a very slow rate $O(n^{-0.11})$. This empirically shows that the sigmoid gating is more sample efficient than the softmax gating.

**References**

[4] D. Dai. DeepSeekMoE: Towards Ultimate Expert Specialization in Mixture-of-Experts Language Models.

[21] H. Nguyen. On least squares estimation in softmax gating mixture of experts.

[34] Jakub Krajewski. Scaling laws for fine-grained mixture of experts.

[35] Xu Owen He. Mixture of A Million Experts.

---

### Decision · Program_Chairs · 2024-09-25

**Decision:**

Accept (poster)

**Comment:**

This work presents a theoretical analysis to examine why sigmoid gating functions exhibit superior performance to softmax gating functions for mixture of experts models. Reviewers appreciated the novelty of the work, acknowledging that sigmoid gating functions are uncommon, and that the effect of selecting different gating functions is a curious topic worth investigating. The authors did an excellent job of responding to reviewer feedback, and while some concerns still remain with the presentation, many of the more technical issues have been resolved. I would strongly recommend that the authors consider making edits (perhaps by including passages from the discussion) to ensure future readers do not adopt the same misunderstandings, and aim to reduce technicality by moving less critical details to the supplementary material. Conditional on such edits, I recommend acceptance.